# Accelerating Relative Entropy Coding
# with Space Partitioning

**Jiajun He**
University of Cambridge
jh2383@cam.ac.uk

**Gergely Flamich**
University of Cambridge
gf332@cam.ac.uk

**José Miguel Hernández-Lobato**
University of Cambridge
jmh233@cam.ac.uk

## Abstract

Relative entropy coding (REC) algorithms encode a random sample following a target distribution $Q$, using a coding distribution $P$ shared between the sender and receiver. Sadly, general REC algorithms suffer from prohibitive encoding times, at least on the order of $2^{D_{\mathrm{KL}}[Q||P]}$, and faster algorithms are limited to very specific settings. This work addresses this issue by introducing a REC scheme utilizing space partitioning to reduce runtime in practical scenarios. We provide theoretical analyses of our method and demonstrate its effectiveness with both toy examples and practical applications. Notably, our method successfully handles REC tasks with $D_{\mathrm{KL}}[Q||P]$ about three times greater than what previous methods can manage, and reduces the bitrate by approximately 5-15% in VAE-based lossless compression on MNIST and INR-based lossy compression on CIFAR-10, compared to previous methods, significantly improving the practicality of REC for neural compression.

## 1   Introduction

Let's consider a two-party communication problem where the sender wants to transmit some data $\mathbf{X}$ to the receiver. A widely used approach is transform coding (Ballé et al., 2020), where $\mathbf{X}$ is first transformed to a discrete variable $\mathbf{Z}$ and entropy coded to achieve an optimal codelength on average. However, directly finding a discrete $\mathbf{Z}$ is difficult in many scenarios. For example, in lossy image compression, $\mathbf{X}$ represents some image, and $\mathbf{Z}$ represents the latent embedding output by an encoder network in a model akin to the variational auto-encoder (Kingma and Welling, 2013). A common solution to obtain a discrete $\mathbf{Z}$ is to first learn a continuous variable and quantize it (Ballé et al., 2017).

However, perhaps surprisingly, there is also a way to directly handle a continuous $\mathbf{Z}$ in this pipeline. Specifically, instead of a deterministic value, the sender transmits a *random* sample following a posterior $\mathbf{Z} \sim Q_{\mathbf{Z}|\mathbf{X}}$[1]. These algorithms are referred to as *relative entropy coding* (REC, Flamich et al., 2020), and also known as *channel simulation* or *reverse channel coding* (Theis and Yosri, 2022). Li and El Gamal (2018) showed that the codelength of such an algorithm is upper-bounded by the mutual information[2] between $\mathbf{X}$ and $\mathbf{Z}$ plus some logarithmic and constant overhead:

$$I[\mathbf{X}; \mathbf{Z}] + \log_2(I[\mathbf{X}; \mathbf{Z}] + 1) + \mathcal{O}(1). \tag{1}$$

REC has a clear advantage over quantization: quantization is a non-differentiable operation, while REC directly works for continuous variables and eases the training of some neural compression models, which highly rely on gradient descent. Also, Theis and Agustsson (2021) exemplified that stochastic encoders can be significantly better than their deterministic counterpart if we target realism.

---

[1]To avoid notation overload, we will use $Q$ for the posterior, omitting its dependence on $\mathbf{X}$ unless needed.

[2]Throughout this paper, unless otherwise stated, we use $\log_2$ to calculate the log density ratio, the Kullback-Leibler (KL) divergence $D_{\mathrm{KL}}$, the mutual information $I$ and the Rényi-$\infty$ divergence $D_\infty$. We use upper-case letters (e.g., $Q$ and $P$) to represent probability measures and lower-case letters (e.g., $q$ and $p$) for their densities.

38th Conference on Neural Information Processing Systems (NeurIPS 2024).

However, the encoding time of REC algorithms is at least typically on the order of $2^{D_{\mathrm{KL}}[Q \parallel P]}$, which is prohibitively long in practice.[3] While there are several works on accelerating REC (Flamich et al., 2022; Flamich and Theis, 2023; Flamich et al., 2024), so far they only work on highly limited problems. In fact, in practice, the common option to employ REC is to segment $\mathbf{Z}$ into the concatenation of independent blocks, denoted as $\mathbf{Z} = [\mathbf{Z}_1, \mathbf{Z}_2, \cdots, \mathbf{Z}_K]$, where the coding cost of each block $\mathbf{Z}_k$ is approximately $\kappa$ bits. This strategy reduces the runtime to $\mathcal{O}(K 2^\kappa)$. However, by Equation (1), the logarithmic and constant overhead only becomes negligible if $I[\mathbf{X}; \mathbf{Z}_i]$ is sufficiently large. For example, for the runtime to be feasible, $\kappa$ is set between 16 to 20 in Havasi et al. (2019); Guo et al. (2024); He et al. (2023). In this case, the overhead will typically constitute 40% to 50% of the total mutual information, resulting in sub-optimal compression performance.

Our work's primary aim is to reduce the complexity of REC runtime for more practical settings. Specifically, our contributions are

- We propose a faster REC framework based on space partitioning. Equivalently, our method can be viewed as introducing a search heuristic to a standard REC algorithm, which can significantly reduce the algorithm's runtime when chosen appropriately. Furthermore, we provide theoretical analysis, showing that our method achieves a close codelength to Equation (1) with an extra cost $\epsilon$ that is negligible for some commonly used distributions in neural compression.

- We show that, interestingly, using different space partitioning and different search heuristics only influences the runtime but not the (upper bound on) codelength. Following this, we discuss two cases: 1) encoding exact samples from the target and 2) encoding approximate samples to further reduce the computational complexity.

- We draw attention to a hitherto unused fact for designing fast, practical relative entropy coding schemes: when the sender wishes to communicate vector-valued random variate $\mathbf{Z} \mid \mathbf{X}$ to the receiver, we may assume that they share knowledge of the *dimension-wise mutual information* $I[Z_i; \mathbf{X}]$ for each dimension $i$, as opposed to just the total $I[\mathbf{Z}; \mathbf{X}]$. Incorporating this information into our space partitioning scheme allows us to construct faster relative entropy coding algorithms.

- We conduct experiments on synthetic examples and neural codecs, including a VAE-based lossless codec on MNIST (LeCun and Cortes, 1998) and INR-based lossy codecs on CIFAR-10 (Krizhevsky et al., 2009). We demonstrate that our method can handle blocks with larger $\kappa$ using much fewer samples and reduces the bitrate by approximately 5-15% compared to previous methods.

## 2  Preliminary

In this paper, we focus on accelerating the *Poisson functional representation* (PFR; Li and El Gamal, 2018) and *ordered random coding* (ORC; Theis and Yosri, 2022), and hence we discuss these in detail below. However, we note that our space partitioning scheme is also applicable to other relative entropy coding (REC) algorithms (Flamich and Theis, 2023; Flamich et al., 2024; Flamich, 2024).

**Relative entropy coding (REC).** Consider a two-party communication problem, where the sender wants to transmit some data $\mathbf{X}$ to the receiver. However, instead of encoding $\mathbf{X}$ directly, the sender first transforms $\mathbf{X}$ into a representation $\mathbf{Z}$ that they encode. In REC, we allow $\mathbf{Z}$ to be stochastic with $\mathbf{Z} \sim Q_{\mathbf{Z}|\mathbf{X}}$. Then, the goal of REC is to encode a *single*, *random* realization $\mathbf{Z} \sim Q_{\mathbf{Z}|\mathbf{X}}$ with the assumption that the sender and receiver share a coding distribution $P$ and have access to common randomness $S$. In practice, the latter can be achieved by a shared pseudo-random number generator (PRNG) and seed. Given these assumptions, the optimal coding cost is given by $\mathbb{H}[\mathbf{Z} \mid S]$, as the code cannot depend on $\mathbf{X}$ since the receiver doesn't know it. Surprisingly, $\mathbf{Z}$ can be encoded very efficiently, as Li and El Gamal (2018) show that

$$I[\mathbf{X}; \mathbf{Z}] \leqslant \mathbb{H}[\mathbf{Z} \mid S] \leqslant I[\mathbf{X}; \mathbf{Z}] + \log_2(I[\mathbf{X}; \mathbf{Z}] + 1) + \mathcal{O}(1). \tag{2}$$

Note that $I[\mathbf{X}; \mathbf{Z}]$ and hence $\mathbb{H}[\mathbf{Z} \mid S]$ can be finite even if $\mathbf{Z}$ is continuous and $\mathbb{H}[\mathbf{Z}]$ is infinite. Next, we describe a concrete scheme with which we can encode $\mathbf{Z}$ at such an efficiency.

**Poisson functional representation (PFR).** To encode a sample from the target distribution $Q$ with density $q$ using the coding distribution $P$ with density $p$, PFR (Li and El Gamal, 2018) starts by

---

[3]We note that REC's decoding is very fast, so in this paper, runtime will refer exclusively to encoding time.

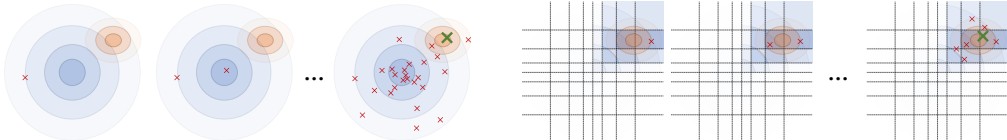

(a) Procedure of standard REC algorithm.    (b) Procedure of REC with space partitioning.

Figure 1: An illustrative comparison between the standard REC algorithm and REC with space partitioning. We illustrate the prior $P$'s density in blue and $Q$'s density in orange. (a) In a standard REC algorithm, we may draw numerous samples (colored in red) before identifying one that aligns well with $Q$ (colored in green). The majority of these samples do not directly contribute to the desired result. (b) In the method we propose, we first divide the search space into smaller grids and then reweight each grid. This amounts to adjusting the prior $P$ to a search heuristic $P'$, which can align better with $Q$. The samples from $P'$ will thus be more relevant to $Q$, potentially reducing the runtime.

drawing a random sequence $\mathbf{Z}_1, \mathbf{Z}_2, \cdots$ from $P$ using the public random state $S$. Furthermore, the sender draws a sequence of random times $T_1, T_2, \cdots$ as follows:

$$T_0 = 0, \quad T_n \leftarrow T_{n-1} + \Delta T_n, \quad \Delta T_n \sim \text{Exp}(1), \quad n = 1, 2, \cdots \tag{3}$$

Next, letting $r = q/p$ be the density ratio, the sender calculates $\tau_n \leftarrow T_n/r(\mathbf{Z}_n)$ for each sample and returns $N^* \leftarrow \arg\min_{i \in \mathbb{N}}\{\tau_i\}$. Using the theory of Poisson processes, it can be shown that $\mathbf{Z}_{N^*} \sim Q$ as desired (Maddison, 2016). Additionally, while the minimum is taken over all positive integers, in practice, $N^*$ can be found in finite steps if $r$ is bounded. In fact, in expectation, PFR will halt after $2^{D_\infty[Q \| P]}$ steps (Maddison, 2016). We summarize this process in Algorithm 1.

**Ordered random coding (ORC).** Unfortunately, PFR's random runtime can be a significant drawback. In practice, we may want to set a limit on the number of iterations to ensure a consistent and manageable runtime at the cost of some bias in the encoded sample. To this end, Theis and Yosri (2022) proposed ordered random coding (ORC). Specifically, in ORC with $N$ candidates, rather than calculating $T_n$ by Equation (3), we first draw $N$ i.i.d. sample from $\text{Exp}(1)$, and sort them in ascending order as $\tilde{T}'_1 \leqslant \tilde{T}'_2 \leqslant \cdots \leqslant \tilde{T}'_N$. We then set $T_n = \tilde{T}'_n$ for each $n = 1, 2, \cdots, N$. In practice, instead of generating the $T_n$-s in $\mathcal{O}(N \log N)$ time by sorting, Theis and Yosri (2022) suggest an iterative procedure similar to Equation (3) with $\mathcal{O}(N)$ time complexity:

$$T_0 = 0, \quad T_n \leftarrow T_{n-1} + {}^N\!/\!(N - n + 1)\Delta T_n, \quad \Delta T_n \sim \text{Exp}(1), \quad n = 1, 2, \cdots, N \tag{4}$$

## 3 Relative Entropy Coding with Space Partitioning

In this section, we describe our proposed algorithm and analyze its codelength. To motivate our method, recall that in PFR, the sender draws a random sequence from $P$, and examines each sample's density ratio. We can interpret this process as a random search in $P$'s support, aiming to find a point that has a relatively high density ratio between $Q$ and $P$. However, when $Q$ concentrates only within a small region of $P$'s support, most of the search does not contribute to the final outcome. Thus, *can we instead quickly navigate the search towards the region where $Q$ is concentrated?*

For one-dimensional distributions $Q$ and $P$, the answer is affirmative, as we can perform a branch-and-bound search by partitioning the 1D space on the fly (Maddison et al., 2014; Flamich et al., 2022). Unfortunately, it is unclear how to generalize these adaptive partitioning strategies to spaces with dimension greater than one. Instead, in Section 3.1, we propose to partition the space in advance according to a rule that the sender and receiver agree on such that we can carry out the search fast enough in practical problems and retain an efficient codelength.

### 3.1 Coding Scheme

Given a shared coding distribution $P$ and a target $Q$, our algorithm proceeds as follows:

1. The sender and receiver agree on a partition of $P$'s support consisting of $J$ bins $\{B_1, \cdots, B_J\}$ with equal probability mass, i.e. $P(B_j) = {}^1\!/\!J$ for $j = 1, \ldots, J$. As we will discuss later, these

**Algorithm 1** Encoding of standard PFR

**Input:** $Q$, $P$ and a random state $S$.
**Output:** Sample index $N^*$.
  # initialize:
  $\tau^* \leftarrow \infty, t_0 \leftarrow 0, N^* \leftarrow 0$ .
  $r_{\max} \leftarrow \sup_{\mathbf{z}} \left\{ \frac{q(\mathbf{z})}{p(\mathbf{z})} \right\}$ .

  # run PFR:
  **for** $n = 1, 2, \cdots$ **do**
    Sample $\Delta t_n \sim \mathrm{Exp}(1); t_n \leftarrow t_{n-1} + \Delta t_n$.
    # simulate a sample with PRNG:

    $\mathbf{z}_n \leftarrow \mathrm{PRNG}(P, S, n)$[4]
    # update $\tau^*$:
    $\tau_n \leftarrow t_n \cdot p(\mathbf{z}_n)/q(\mathbf{z}_n)$.
    **if** $\tau_n \leqslant \tau^*$ **then**
      $\tau^* \leftarrow \tau_n, N^* \leftarrow n$.
    **end if**
    # check stopping criterion:
    **if** $t_n/r_{\max} > \tau^*$ **then**
      **break**
    **end if**
  **end for**

---

**Algorithm 2** PFR with Space Partitioning

**Input:** $Q$, $P$, and random states $\{S_j\}_{j=1}^{J}$.
**Output:** Bin index $j^*$, local sample index $\tilde{n}^*$.
  # initialize:
  $\tau^* \leftarrow \infty, t_0 \leftarrow 0, j^* \leftarrow 0, \tilde{n}^* \leftarrow 0$.
  Partition space in $J$ "bins", s.t. $P(B_j) = 1/J$.
  Select categorical distribution $\pi$.
  $\tilde{n}_j \leftarrow 0$ for $j = 1, 2, \cdots, J$.
  $r'_{\max} \leftarrow \max_{j=1,\cdots,J} \left\{ \sup_{\mathbf{z} \in B_j} \left\{ \frac{q(\mathbf{z})}{p(\mathbf{z})} \frac{P(B_j)}{\pi(j)} \right\} \right\}$.
  # run PFR:
  **for** $n = 1, 2, \cdots$ **do**
    Sample $\Delta t_n \sim \mathrm{Exp}(1); t_n \leftarrow t_{n-1} + \Delta t_n$.
    # simulate a sample with PRNG:
    $j_n \sim \pi; \tilde{n}_{j_n} \leftarrow \tilde{n}_{j_n} + 1$.
    $\mathbf{z}_n \leftarrow \mathrm{PRNG}(P|_{B_{j_n}}, S_{j_n}, \tilde{n}_{j_n})$.
    # update $\tau^*$:
    $\tau_n \leftarrow J \cdot \pi(j_n) \cdot t_n \cdot p(\mathbf{z}_n)/q(\mathbf{z}_n)$.
    **if** $\tau_n \leqslant \tau^*$ **then**
      $\tau^* \leftarrow \tau_n, j^* \leftarrow j_n, \tilde{n}^* \leftarrow \tilde{n}_{j_n}$.
    **end if**
    # check stopping criterion:
    **if** $t_n/r'_{\max} > \tau^*$ **then**
      **break**
    **end if**
  **end for**

---

bins can overlap, but to aid understanding for now, it may be helpful to imagine the space is partitioned by non-overlapping bins.

2. According to the target distribution $Q$, the sender selects a categorical distribution for the bins, with event probabilities defined as $\pi(j)$ for $j = 1, 2, \cdots, J$. We can view this categorical distribution as a reweighting of each bin to adjust the coding distribution $P$. The coding distribution adjusted by this reweighting can be better aligned with $Q$. We will discuss the choice of $\pi$ later.

3. Then, the sender starts to draw and examine samples iteratively, similar to the PFR algorithm. However, instead of drawing samples directly from $P$, the sender first samples a bin from $\pi$ and then draws a sample from the prior restricted to this bin. Specifically, at each iteration $n$:

   (a) the sender first draws a bin index, $j_n$, according to the distribution $\pi$;
   (b) the sender then draws a sample $\mathbf{Z}_n \sim P|_{B_{j_n}}$. This sample is generated with the random state $S_{j_n}$ associated with this bin. It's critical that each bin has a unique random state to ensure different bins have different random sample sequences. In practice, this can be easily achieved by setting the PRNG's seed of the bin $B_j$ to $j$, for $j = 1, 2, \cdots, J$. Note that given these random states, the value of $\mathbf{Z}_n$ is uniquely determined by the bin index $j_n$ and its sample index within this bin, which we denote by $\tilde{n}_{j_n}$. For the sake of clarity, we will call $n$ the *global sample index* and $\tilde{n}_{j_n}$ the *local sample index* hereafter;
   (c) the sender examines the sample by calculating $\tau_n$:

$$\tau_n \leftarrow T_n/r', \quad r' \coloneqq q(\mathbf{Z}_n)/p'_n, \quad p'_n \coloneqq \pi(j_n)p(\mathbf{Z}_n)/P(B_{j_n}) = J \cdot \pi(j_n) \cdot p(\mathbf{Z}_n), \quad (5)$$

   and keeps track of $N^* \leftarrow \arg\min_{i=1,2,\cdots,n}\{\tau_i\}$. Here, $T_n$ is also obtained by Equation (3);
   (d) finally, the sender checks the following stopping criterion:

$$T_n/r'_{\max} > \tau^*, \quad r'_{\max} \coloneqq \max_{j=1,2,\cdots,J} \left\{ \sup_{\mathbf{z} \in B_j} \frac{q(\mathbf{z})P(B_j)}{p(\mathbf{z})\pi(j)} \right\} \quad (6)$$

   If the criterion is met, halt and return $j_{N*}$ and $\tilde{n}_{N*}$; otherwise, proceed to the next iteration.

---

[4]$\mathrm{PRNG}(P, S, n)$ means simulating the $n$-th sample in the random sequence from $P$ by PRNG with seed $S$.

We detail the above coding scheme in Algorithm 2. As a comparison, we describe the standard PFR algorithm in Algorithm 1, and highlight their difference in red. We also illustrate these two algorithms from a high-level point of view in Figure 1. It is easy to verify that Algorithm 2 is equivalent to running standard PFR with an adjusted prior $P'$, whose density is defined as

$$p'(\mathbf{z}) = \sum_{j=1}^{J} \mathbb{1}\{\mathbf{z} \in B_j\} \frac{\pi(j)p(\mathbf{z})}{P(B_j)}. \tag{7}$$

Therefore, the encoded sample is guaranteed to be $Q$-distributed. By choosing a sensible $\pi$, the adjusted prior $P'$ can be more closely aligned with $Q$ than the original $P$, potentially decreasing the runtime. From this point forward, we will refer to $P'$ as the *search heuristic*.

Note that the receiver does not need to be aware of $\pi$ to decode the sample. Instead, after receiving the bin index $j_{N*}$, the receiver can construct the random sequence from $P|_{B_{j_{N*}}}$ with seed $S_{j_{N*}}$. Then, taking the $\tilde{n}_{N*}$-th sample in this sequence, the receiver successfully retrieves the desired sample.

### 3.2 Codelength of the two-part code

In our proposed algorithm, the sender needs to transmit a two-part code that includes both the bin index and the local sample index. This is distinct from the standard PFR, where the sender transmits only the sample index, hence potentially raising concerns about the codelength. Indeed, we find the two-part code can introduce an extra cost, as outlined in the following theorem:

**Theorem 3.1.** *Let a pair of correlated random variables* $\mathbf{X}, \mathbf{Z} \sim P_{\mathbf{X},\mathbf{Z}}$ *be given. Assume we perform relative entropy coding using Algorithm 2 and let $j^*$ denote the bin index and $\tilde{n}^*$ the local sample index returned by the algorithm. Then, the entropy of the two-part code is bounded by*

$$\mathbb{H}[j^*, \tilde{n}^*] \leqslant I[\mathbf{X}; \mathbf{Z}] + \mathbb{E}_{\mathbf{X}}[\epsilon] + \log_2(I[\mathbf{X}; \mathbf{Z}] - \log_2 J + \mathbb{E}_{\mathbf{X}}[\epsilon] + 1) + 4, \tag{8}$$

$$\text{where} \quad \epsilon = \mathbb{E}_{\mathbf{z} \sim Q_{\mathbf{Z}|\mathbf{X}}} \left[ \max \left\{ 0, \log_2 J - \log_2 \frac{q(\mathbf{z})}{p(\mathbf{z})} \right\} \right]. \tag{9}$$

We prove Theorem 3.1 in Appendix C.1. Note, that when $I[\mathbf{X}; \mathbf{Z}]$ is sufficiently large, the term $\log_2(I[\mathbf{X}; \mathbf{Z}] - \log_2 J + \mathbb{E}_{\mathbf{X}}[\epsilon] + 1) + 4$ will be negligible. However, without further information on $Q$ and $P$, it is difficult to assert how small $\mathbb{E}_{\mathbf{X}}[\epsilon]$ is. We can view $\epsilon$ as the extra cost introduced by the space partitioning algorithm. Fortunately, for commonly used distributions in neural compression, including Uniform and Gaussian, we have the following conclusion under reasonable assumptions:

**Proposition 3.2** (Bound of $\epsilon$ for Uniform and Gaussian). *Assume setting* $J \leqslant 2^{D_{\mathrm{KL}}[Q \,\|\, P]}$ *when running Algorithm 2 for each $Q_{\mathbf{Z}|\mathbf{X}}$. Then, for Uniform $Q$ and $P$, if $Q \ll P$ (i.e., $Q$ is absolute continuous w.r.t $P$), we have $\epsilon = 0$; for factorized Gaussian $Q$ and $P$, if $Q$ has smaller variance than $P$ along each axis, we have* $\epsilon \leqslant 0.849 \sqrt{D_{\mathrm{KL}}[Q \,\|\, P]}$, *and* $\mathbb{E}_{\mathbf{X}}[\epsilon] \leqslant 0.849 \sqrt{I[\mathbf{X}; \mathbf{Z}]}$.

We prove Proposition 3.2 in Appendix C.2. Also, we highlight that the conclusion for Gaussian in Proposition 3.2 is derived by considering the worst case when $P$ has the same variance as $Q$ along *all* dimensions. In practice, this worst case can barely happen since in neural compression $P$ represents the prior, and $Q$ represents the posterior, which will be more concentrated than the prior. Empirically, we find this $\epsilon$-cost yields no visible influence on the codelength (e.g., Figure 2b in Section 5).

### 3.3 Generality of our Space Partitioning Algorithm

We highlight that the conclusions in Section 3.2 are independent of the partitioning strategy and $\pi$. This allows us to use different $\pi$ without the need to revisit the bound of the codelength.

More interestingly, we do not need to partition the space with non-overlapping bins. This is because the density of $P'$, as stated in Equation (7), can be directly interpreted as a mixture of priors with $J$ components, where $\pi(j)$ represents the mixture weights. There are no restrictions preventing this mixture model from having overlapping components. We provide more explanation and discussion on overlapping components in Appendix B.1. As an extreme case, we can even have entirely overlapping bins. Notably, this scenario coincides with the "parallel threads" version of REC proposed by Flamich (2024). Concretely, Flamich (2024) proposed to initiate several threads for a single REC task and run them in parallel on various machines, thereby decreasing the runtime.

Furthermore, the concept of space partitioning and its codelength, as stated in Theorem 3.1, extends beyond the scope of Poisson functional representation algorithms. Here, we state the codelength by applying our proposed space partitioning method to greedy Poisson rejection sampling (GPRS) (Flamich, 2024) and leave its application to other REC algorithms for future work.

**Theorem 3.3.** *Let a pair of correlated random variables* $\mathbf{X}, \mathbf{Z} \sim P_{\mathbf{X}, \mathbf{Z}}$ *be given. Assume we perform relative entropy coding using GPRS with space partitioning and let $j^*$ denote the bin index and $\tilde{n}^*$ the local sample index returned by the algorithm, and $\epsilon$ be as in Equation (9). Then,*

$$\mathbb{H}[j^*, \tilde{n}^*] \leqslant I[\mathbf{X}; \mathbf{Z}] + \mathbb{E}_{\mathbf{X}}[\epsilon] + \log_2(I[\mathbf{X}; \mathbf{Z}] - \log_2 J + \mathbb{E}_{\mathbf{X}}[\epsilon] + 1) + 6. \tag{10}$$

We prove Theorem 3.3 in Appendix C.3. Additionally, since we can view the "parallel threads" version of GPRS as a special case of our proposed method, Theorem 3.3, and hence Proposition 3.2, offer alternative bounds for Theorem 3.5 in Flamich (2024).

## 4 Exemplifying the choice of Partitioning Strategy and $\pi$

In the above sections, we do not specify the partitioning strategy and $\pi$. In this section, we will exemplify their choices. To keep our discussion simple, we take the assumption that both $Q$ and $P$ are fully-factorized distributions. This covers a majority of neural compression applications with relative entropy coding, including both VAE-based (Flamich et al., 2020) and INR-based codecs (Guo et al., 2024; He et al., 2023). Under this assumption, we adopt a simple partitioning strategy to split space with axis-aligned grids, which allows us to draw samples and evaluate the density ratio easily.

Now, let's focus on $\pi$. We consider two scenarios: 1) the sender aims to encode a sample that exactly follows $Q$, and 2) the sender encodes a sample following $Q$ approximately for a more manageable computational cost. As we will see, the optimal choice of $\pi$ differs in these cases.

We note that the latter garners more interest in practical neural compression. This is because we can always construct an example where the standard PFR algorithm does not terminate while the first sample from the prior already has little bias. As an example, take $q(z) = \mathcal{N}(z|0.001, \sigma^2)$, and $p(z) = \mathcal{N}(z|0, 1)$. When $\sigma \nearrow 1$ (approaching 1 from below), the expected runtime for PFR diverges: $2^{D_\infty[Q||P]} \to \infty$. Unfortunately, our proposed space partitioning approach can do little in this case. This is because, for a small $\epsilon$-cost, as stated in Proposition 3.2, the number of partitions $J$ should be less than $2^{D_{\mathrm{KL}}[Q \parallel P]}$, which reduces to 1 when $\sigma \nearrow 1$. For multi-dimensional (factorized) $Q$ and $P$, this issue will occur whenever it arises in *any single* dimension. Making things even worse, this example is pervasive in neural compression due to numerical inaccuracies. Therefore, limiting the number of candidate samples is more practical than running the PFR algorithm until the stopping criterion is met. Consequently, we will mainly study the non-exact case in the following, but for the sake of completeness, we first discuss the exact scenario.

### 4.1 Exact Sampler

When encoding a sample following $Q$ exactly, we hope to minimize the expected runtime by adjusting the search heuristic $P'$. Since we can view our proposed algorithm as running standard PFR with $P'$, its expected runtime is $2^{D_\infty[Q||P']}$. Thus, we have the following constrained optimization problem:

$$\underset{\pi}{\arg\min} \left\{ \max_{j=1,2,\cdots,J} \left[ \sup_{\mathbf{z} \in B_j} \frac{q(\mathbf{z})P(B_j)}{p(\mathbf{z})\pi(j)} \right] \right\}, \quad \text{subject to } \sum_{j=1}^{J} \pi(j) = 1. \tag{11}$$

The solution turns out to be intuitive:

$$\pi(j) \propto \sup_{\mathbf{z} \in B_j} \frac{q(\mathbf{z})P(B_j)}{p(\mathbf{z})} \propto \sup_{\mathbf{z} \in B_j} \frac{q(\mathbf{z})}{p(\mathbf{z})} . \tag{12}$$

However, sampling from this $\pi$ is generally challenging. Fortunately, if $Q$ and $P$ are both factorized and we partition the space with axis-aligned grids, then $\pi$ factorizes dimensionwise into a product of categorical distributions. Also, this choice of $\pi$ simplifies the evaluation of the stopping criterion in Equation (6). Specifically, $r'_{\max}$ simplifies to $r'_{\max} = Z \cdot J$, where $Z = \sum_{j=1}^{J} \pi(j)$ denotes the normalization constant. This constant is shared in both the maximum density ratio $r'_{\max}$ and the density ratio for individual samples. We thus can omit it when evaluating the stopping criterion. We detail the procedure in Algorithm 3 in Appendix A.1.

## 4.2 Non-exact Sampler

If we only need to encode a sample following $Q$ approximately, we can apply our space partitioning strategy to ordered random coding (ORC). In this case, we hope to reduce the bias with the same sample size or reduce the sample size for the same bias. To determine the optimal choice of $\pi$, we state the following corollary on ORC's bias. This is a corollary of Theis and Yosri (2022, Lemma D.1) and Chatterjee and Diaconis (2018, Theorem 1.2). We present the proof in Appendix C.4.

**Corollary 4.1** (Biasness of sample encoded by ORC). *Given a target distribution $Q$ and a prior distribution $P$, run ordered random coding (ORC) to encode a sample. Let $\tilde{Q}$ be the distribution of encoded samples. If the number of candidates is $N = 2^{D_{\mathrm{KL}}[Q \parallel P]+t}$ for some $t \geqslant 0$, then*

$$D_{TV}[\tilde{Q}, Q] \leqslant 4 \left( 2^{-t/4} + 2\sqrt{\mathbb{P}_{\mathbf{z} \sim Q} \left( \log \frac{q(\mathbf{z})}{p(\mathbf{z})} \geqslant D_{\mathrm{KL}}[Q \parallel P] + \frac{t}{2} \right)} \right)^{1/2}. \tag{13}$$

*Conversely, supposing that $N = 2^{D_{\mathrm{KL}}[Q \parallel P]-t}$ for some $t \geqslant 0$, then*

$$D_{TV}[\tilde{Q}, Q] \geqslant 1 - 2^{-t/2} - \mathbb{P}_{\mathbf{z} \sim Q} \left( \log \frac{q(\mathbf{z})}{p(\mathbf{z})} \leqslant D_{\mathrm{KL}}[Q \parallel P] - \frac{t}{2} \right). \tag{14}$$

This corollary tells us that when running ORC with target $Q$ and prior $P$, if the density ratio between $Q$ and $P$ is well concentrated around its expectation, *choosing $N \approx 2^{D_{\mathrm{KL}}[Q \parallel P]}$ candidates is both sufficient and necessary to encode a low-bias sample in terms of total variation (TV) distance*. Recall that our proposed algorithm can be viewed as running ORC with the search heuristic $P'$ as the prior. We, therefore, want to choose $\pi$ to minimize the KL-divergence between the target $Q$ and the search heuristic $P'$. The optimal $\pi$ turns out to be surprisingly simple:

$$\pi(j) = Q(B_j), j = 1, 2, \cdots, J. \tag{15}$$

This is because $D_{\mathrm{KL}}[Q \parallel P'] = D_{\mathrm{KL}}[Q \parallel P] - \sum_{j=1}^{J} Q(B_j) \log_2 \pi(j) + \sum_{j=1}^{J} Q(B_j) \log_2 P(B_j)$, and the cross-entropy $(-\sum_{j}^{J} Q(B_j) \log_2 \pi(j))$ takes its minimum when $\pi$ matches $Q$. In practice, to sample from this categorical distribution, we can simply draw $\mathbf{z} \sim Q$ and find the bin it belongs to. However, choosing $\pi(j) = Q(B_j)$ complicates the calculation of $r'_{\max}$ in Equation (6). Fortunately, in ORC, we do not need to check the stopping criterion, so this complication does not pose an issue. We formalize this new ORC algorithm in Algorithm 4 in Appendix A.2.

Algorithm 4 still leaves three questions unanswered: first, we need to *determine the number of partitions $J$*. As a finer partition allows better alignment between $Q$ and $P'$, we pick $J = 2^{\lfloor D_{\mathrm{KL}}[Q \parallel P] \rfloor}$, the maximum value for which Proposition 3.2 provides a bound on the extra coding cost $\epsilon$. Note that this will require the receiver to know $\lfloor D_{\mathrm{KL}}[Q \parallel P] \rfloor$. As we will demonstrate in Section 5, in practice, we can achieve this by either encoding the KL using negligible bits (e.g., in neural compression with VAE) or enforcing the KL budget during optimization (e.g., in neural compression with INRs).

Second, as we partition the space using axis-aligned grids, we need to *determine the number of bins assigned to each axis*. To explain why this choice matters, we consider an example where $Q$ and $P$ share the same marginal distribution along a specific axis, and the space is partitioned solely by dividing this axis; in this case, we have $P' \equiv P$, leading to no improvement in runtime. Fortunately, in most neural compression applications, the sender and receiver can also have access to an estimation of the mutual information $I_d$ along each axis $d$ from the training set. If the mutual information is *non-zero* along one axis, on average, $Q$ and $P$ will *not* have the same marginal distribution along this axis. Based on this observation, we suggest partitioning the $d$-th axis into approximately $2^{n_d}$ intervals where $n_d = D_{\mathrm{KL}}[Q \parallel P] \cdot I_d / \sum_{d'=1}^{D} I_{d'}$; see Appendix B.2 for further discussion, including the derivation, a toy example illustration, and ablation studies on this.

Third, we *determine how many candidate samples we need to draw* from $P'$ to ensure the encoded sample has low bias. Recall that our proposed algorithm can be viewed as running ORC with $P'$ as the prior. We thus require a sample size $\approx 2^{D_{\mathrm{KL}}[Q \parallel P']}$ according to Corollary 4.1. When the total number of partitions $J = 2^{D_{\mathrm{KL}}[Q \parallel P]}$, we find $D_{\mathrm{KL}}[Q \parallel P'] = -\sum_{j=1}^{J} Q(B_j) \log_2 Q(B_j)$. We do not have an analytical form for this value, but we can estimate it by samples from $Q$. In fact, we empirically find a small number of samples to be sufficient for an accurate estimator.

**Why choose the TV distance as the measure of approximation error in Corollary 4.1?** Indeed, TV distance is not the only possible metric and the choice should align with the intended application. We choose total variation as it fits well with the task in our experiments: the one-shot nature of data compression for human consumption. We will explain this in two parts:

1. *Control of the TV distance in the latent space implies control in data space:* naturally, we wish to assess reconstruction quality in data space, but we use REC only to encode latent variables from which we reconstruct the data. However, since the total variation satisfies the data processing and triangle inequalities, if our generative model approximates the true data distribution with $\delta$ total variation error and we use an $\epsilon$-approximate REC algorithm, then encoding the latents incurs no more than $\epsilon + \delta$ total variation error in the data space (Flamich and Wells, 2024).

2. *TV distance captures a reasonable notion of realism.* Imagine two distributions $Q$ (e.g., the ground truth data distribution) and $\tilde{Q}$ (e.g., the data distribution learned by our compressor). Let $\mathbf{x}_0 \sim Q, \mathbf{x}_1 \sim \tilde{Q}$ and let $B \sim \mathrm{Bern}(1/2)$ be a fair coin toss. Then, the probability that any observer can correctly decide which distribution $\mathbf{z}_B$ was sampled from, i.e., correctly predict the value of $B$ given $\mathbf{x}_B$, is at most $1/2 \cdot (1 + D_{\mathrm{TV}}[\tilde{Q}, Q])$ (Nielsen, 2013; Blau and Michaeli, 2018). Thus, in the context of approximate REC, the TV distance bounds the accuracy with which any observer can tell the compressed and reconstructed data apart from the original.

## 5 Experiments

We now verify our proposed algorithm with three different experiments, including synthetic toy examples, lossless compression on MNIST with VAE, and lossy compression on CIFAR-10 with INRs. We include details of the experiment setups in Appendix D.

**Toy Experiments.** We explore the effectiveness of our algorithm on 5D synthetic Gaussian examples. We run PFR with space partitioning (Algorithm 3) to encode exact samples and ORC with space partitioning (Algorithm 4) for approximate samples. We show the results in Figure 2 and Figure 3, and also include standard PFR and ORC for comparison. We can see that our space partitioning algorithm reduces the runtime by up to three orders of magnitude while maintaining codelength when encoding exact samples, and requires a much smaller sample size to achieve a certain bias (quantified by maximum mean discrepancy (MMD, Smola et al., 2007)) when encoding approximate samples.

**Lossless Compression on MNIST with VAE.** As a further proof of concept, we apply our methods to losslessly compress MNIST images (LeCun and Cortes, 1998). We train a VAE following Flamich et al. (2024), employ REC (specifically, ORC with our proposed space partitioning) to encode the latent embeddings, and entropy-encode the image. The KL divergence of entire latent embeddings averages over 90 bits. Unfortunately, even after employing our proposed space partitioning algorithm, the new KL divergence $D_{\mathrm{KL}}[Q \parallel P']$ exceeds our manageable size. We hence randomly divide the latent dimensions into smaller blocks. Recall that, in our proposed approach, the sender and receiver need to share $\lfloor D_{\mathrm{KL}}[Q \parallel P] \rfloor$ so that they can partition the space into the same $J = 2^{\lfloor D_{\mathrm{KL}}[Q \parallel P] \rfloor}$ bins. However, this value varies across different images. Therefore, we estimate the distribution of $\lfloor D_{\mathrm{KL}}[Q \parallel P] \rfloor$ for each block from the training set, and entropy code it for each test image.

We evaluate the performance achieved by $\{2, 4\}$ blocks and different sample sizes in Table 1, with the theoretically optimal codelength and the results by GPRS, a REC algorithm that is faster in 1D space but incurs coding overhead dimension-wise (Flamich, 2024). Notably, our algorithm's codelength is only 2% worse than the theoretically optimal result and about 6% better than GPRS's. We also investigate the reasons for overhead. Compared to the ELBO, our method incurs overhead in three ways: (a) the cost to encode $\lfloor D_{\mathrm{KL}}[Q \parallel P] \rfloor$; (b) the overhead from encoding the latent embeddings by REC; and (c) the overhead when encoding the target image, which arises from the bias in encoding the latent embeddings, as ORC encodes only approximate samples. We can see our algorithm achieves extremely low bias while maintaining a relatively small overhead caused by (a) and (b).

**Lossy Compression on CIFAR-10 with INRs.** We apply our methods to a more practical setting: compressing CIFAR-10 images with RECOMBINER (He et al., 2023), an implicit neural representation (INR)-based codec. The authors of RECOMBINER originally encoded INR weights using a block size of $D_{\mathrm{KL}}[Q \parallel P] = 16$ bits. We empirically find that our proposed method can handle a block size of $D_{\mathrm{KL}}[Q \parallel P] = 48$ bits while maintaining $D_{\mathrm{KL}}[Q \parallel P']$ within a manageable range, approximately 12-14 bits. To further reduce the bias of the encoded sample, we opt to use $2^{16}$ samples

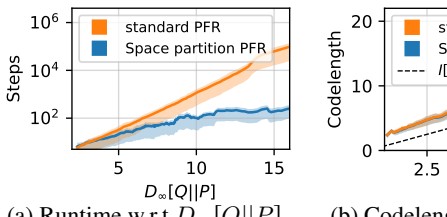

(a) Runtime w.r.t $D_\infty[Q||P]$.

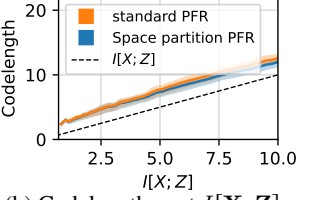

(b) Codelength w.r.t $I[\mathbf{X};\mathbf{Z}]$.

Figure 2: Comparing standard PFR and PFR with our proposed space partitioning algorithm on toy examples. Solid lines and the shadow areas represent the mean and IQR.

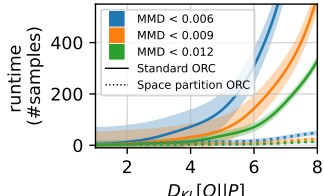

Figure 3: Comparing standard ORC and ORC with our proposed space partitioning algorithm on toy examples.

Table 1: Lossless compression performance on MNIST test set with different REC settings. We include the theoretical results and the results by GPRS (Flamich, 2024) for reference. We repeat each setting 5 times and report the mean and std. **\***: $\bar{\kappa}$ represents the average KL divergence between target $Q$ and the search heuristic $P'$. We estimate it by MC estimator and average across all test images. We empirically find $\bar{\kappa} \approx 14$ when using 2 blocks and $\bar{\kappa} \approx 7$ when using 4 blocks. †: Given that $\bar{\kappa}$ is relatively small, we can employ more than $\lfloor 2^{\bar{\kappa}} \rfloor$ samples to further reduce the bias. We hence report the results achieved by $\lfloor 2^{\bar{\kappa}+2} \rfloor$ samples, which remains manageable in both cases.

| REC SETUPS | | BITRATE | DETAILS OF THE OVERHEAD (TO ELBO) | | |
|---|---|---|---|---|---|
| #BLOCKS | RUNTIME (#SAMPLES / BLOCK ) | BITS PER PIXEL | COST TO ENCODE KL | OVERHEAD IN REC | OVERHEAD FROM BIAS |
| 4 | 1 | $1.6139 \pm 0.0058$ | | 0 | $0.2472 \pm 0.0066$ |
| | 100 | $1.4045 \pm 0.0004$ | | $0.0222 \pm 0.0002$ | $0.0158 \pm 0.0008$ |
| | $\lfloor 2^{\bar{\kappa}} \rfloor$* | $1.4042 \pm 0.0010$ | $0.0217 \pm 0.0001$ | $0.0233 \pm 0.0011$ | $0.0144 \pm 0.0024$ |
| | $\lfloor 2^{\bar{\kappa}+2} \rfloor$† | $1.4012 \pm 0.0006$ | | $0.0293 \pm 0.0008$ | $0.0054 \pm 0.0008$ |
| 2 | 1 | $1.6051 \pm 0.0060$ | | 0 | $0.2463 \pm 0.0063$ |
| | 100 | $1.4089 \pm 0.0005$ | | $0.0105 \pm 0.0001$ | $0.0397 \pm 0.0006$ |
| | $\lfloor 2^{\bar{\kappa}} \rfloor$* | $1.3905 \pm 0.0005$ | $0.0130 \pm 0.0001$ | $0.0259 \pm 0.0010$ | $0.0059 \pm 0.0018$ |
| | $\lfloor 2^{\bar{\kappa}+2} \rfloor$† | $1.3898 \pm 0.0007$ | | $0.0286 \pm 0.0008$ | $0.0024 \pm 0.0010$ |
| Theorical Optimum | | $1.3618 \pm 0.0006$ | 0 | $0.0136 \pm 0.0001$ | 0 |
| GPRS (Flamich, 2024) | | $1.4810 \pm 0.0015$ | 0 | $0.1320 \pm 0.0012$ | $0.0012 \pm 0.0001$ |

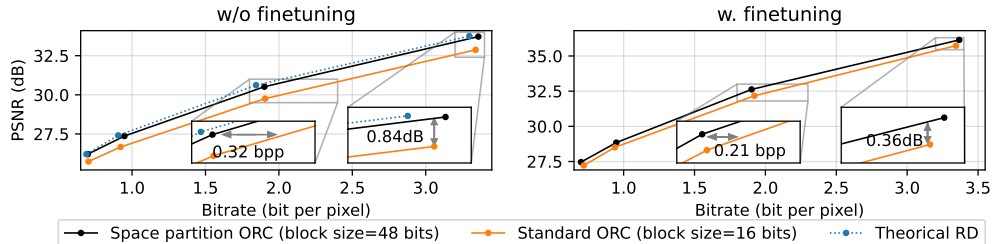

Figure 4: Rate-distortion curve of RECOMBINER by our proposed algorithm and standard ORC. We also provide the theoretical RD curve for an ideal REC algorithm (i.e., assuming we can encode an exact sample in a single block, whose codelength is calculated by Equation (1)). Notably, our method's performance is already very close to this theoretical result.

for each block. Besides, unlike in the VAE case, where the KL for each block varies across different test images, here, we follow He et al. (2023) to enforce the KL of all blocks to be close to 48 bits when training the INR for each test image. This eliminates the need to encode $\lfloor D_{\mathrm{KL}}[Q \parallel P] \rfloor$.

Additionally, He et al. (2023) enhanced their results by fine-tuning $Q$ for the not-yet-compressed weights after encoding each block, which can achieve a more complicated posterior distribution $Q$ in an auto-regressive manner. However, the effectiveness of fine-tuning is closely tied to the number of blocks. As we have reduced the number of blocks in our approach, fine-tuning becomes less effective. Therefore, we present results both with and without fine-tuning in Figure 4. We also present the theoretical RD curve (without fine-tuning) to evaluate how close we are to the theoretical bound.

As we can see, compared with standard ORC using a block size of $D_{\mathrm{KL}}[Q \parallel P] = 16$ bits, our proposed algorithm with the block size of $D_{\mathrm{KL}}[Q \parallel P] = 48$ bits reduces the codelength by approximately 10% with fine-tuning and 15% without. This gain is due to three reasons: 1) when the KL of each block is larger, the $\epsilon$-cost, the algorithmic and constant overhead in Equation (8) becomes negligible; 2) the $-\log_2 J$ term in Equation (8) also reduces the algorithmic overhead in our method's codelength comparing with Equation (1); and 3) as the new KL divergence $D_{\mathrm{KL}}[Q \parallel P']$ is around 12-14 bits and we choose a sample size of $2^{16}$, we eliminate most of the bias in our encoded samples. Moreover, the algorithm's codelength remains within 2% of the theoretical optimal result. As a further comparison, we present the performance of other compression baselines in Figure 12, where we can see that our proposed algorithm makes RECOMBINER more competitive.

## 6    Related Works

There are several efforts to accelerate REC algorithms. Flamich et al. (2022, 2024); Flamich (2024) leveraged the idea of partitioning in 1D space, achieving an impressive runtime in the order of $\mathcal{O}\left(D_\infty[Q \| P]\right)$ or $\mathcal{O}\left(D_{\mathrm{KL}}[Q \parallel P]\right)$. However, these fast approaches only work for 1D distributions. Besides, Flamich and Theis (2023) proposed bits-back quantization (BBQ), which encodes a sample with time complexity linear in dimensionality. However, BBQ assumes $P$ and $Q$ to be uniform distributions in two hypercubes with their edges aligned, which may not be practical in many applications. The algorithm most similar to ours is hybrid coding (Theis and Yosri, 2022), which employs dithered quantization followed by a sampling procedure, an approach that resembles a special case of our proposed algorithm. However, hybrid coding relies on the assumption that the support of $Q$ is contained within a hypercube, restricting its practical applicability.

## 7    Conclusions and Limitations

In this work, we propose a relative entropy coding (REC) scheme based on space partitioning, which significantly reduces the runtime in practical settings. We provide both theoretical analysis and experimental evidence supporting our method. Being among the few successful attempts to accelerate REC for practical settings, we firmly believe that our contributions will broaden the application of REC methods and inspire further research.

However, our proposed algorithm still faces several limitations at the current stage. First, although our method and conclusion apply to general partitions, in practice, we are largely limited to handling axis-aligned grids. Second, in our experiments, we require the mutual information to be factorized per dimension and shared between the sender and receiver, which restricts our algorithm's utility, for example, in non-factorized cases (Theis et al., 2022) or one-shot REC tasks (Havasi et al., 2019). A potential avenue for future research involves employing a mixture of priors to form "partitions" without hard boundaries as discussed in Section 3.3. Additionally, by considering the prior as a convolution of two distributions, we may potentially create infinitely many partitions. However, managing the codelength in such cases may raise new challenges.

## 8    Acknowledgments

We would like to thank Zongyu Guo and Sergio Calvo-Ordoñez for their proofreading and insightful feedback on the manuscript. JH was supported by the University of Cambridge Harding Distinguished Postgraduate Scholars Programme. JMHL and JH acknowledge support from a Turing AI Fellowship under grant EP/V023756/1; GF acknowledges funding from DeepMind.

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

# A Algorithms

## A.1 Practical PFR Algorithm with Space Partitioning

---

**Algorithm 3** Practical encoding procedure of PFR with dimension-wise space partitioning

---

**Input:** Fully-factorized target $Q = Q^{[1]} \times Q^{[2]} \times \cdots \times Q^{[D]}$, shared and fully-factorized coding distribution $P = P^{[1]} \times P^{[2]} \times \cdots \times P^{[D]}$, and shared random states $S_j$ for each partition $j$.
**Output:** Bin index $j^*$, local sample index $\tilde{n}^*$.

  # initialize:
  $\tau^* \leftarrow \infty, t_0 \leftarrow 0, j^* \leftarrow 0, \tilde{n}^* \leftarrow 0$.
  Partition $d$-th dimension into $J^{[d]}$ intervals $B_1^{[d]}, B_2^{[d]}, \cdots, B_{J^{[d]}}^{[d]}$, s.t., $J \leftarrow \prod_{d=1}^{D} J^{[d]}$.
  $\tilde{n}_j \leftarrow 0$ for $j = 1, 2, \cdots, J$.
  $r'_{\max} \leftarrow 1$.                      $\triangleright$ Note that we omit the normalization factor and $J$
  # pre-process for fast sampling along each dimension:
  **for** $d = 1, 2, \cdots, D$ **do**
    **for** $i = 1, 2, \cdots, J^{[d]}$ **do**
      $\pi_i^{[d]} \leftarrow \sup_{z \in B_i^{[d]}} \left( \frac{q^{[d]}(z)}{p^{[d]}(z)} \right)$.
    **end for**
    $Z \leftarrow \sum_{i=1}^{J^{[d]}} \pi_i^{[d]}$.
    Define $\pi^{[d]} = \text{Categorical} \left( \pi_1^{[d]}/Z, \pi_2^{[d]}/Z, \cdots, \pi_{J^{[d]}}^{[d]}/Z \right)$.
  **end for**
  # run PFR:
  **for** $n = 1, 2, \cdots$ **do**
    # sample time:
    Sample $\Delta t_n \sim \text{Exp}(1)$.
    $t_n \leftarrow t_{n-1} + \Delta t_n$.
    # sample a bin by sampling intervals per dimension:
    $j_n \leftarrow 0$.                            $\triangleright$ $j_n$ keeps track of the bin index.
    $\ell_n \leftarrow 1$.                  $\triangleright$ $\ell_n$ keeps track of the likelihood of the bin being sampled.
    **for** $d = 1, 2, \cdots, D$ **do**
      $k \sim \pi^{[d]}$.
      $\ell_n \leftarrow \ell_n \cdot \pi_k^{[d]}$.
      **if** $d = 0$ **then**
        $j_n \leftarrow k$.
      **else**
        $j_n \leftarrow k + j_n \cdot J^{[d-1]}$.
      **end if**
    **end for**
    # simulate a sample with PRNG:
    $\tilde{n}_{j_n} \leftarrow \tilde{n}_{j_n} + 1$.
    $\mathbf{z}_n \leftarrow \text{PRNG}(P|_{B_{j_n}}, S_{j_n}, \tilde{n}_{j_n})$.
        $\triangleright$ by inverse transform sampling when both of the partitioning and $P$ are axis-aligned.
    # update $\tau^*$:
    $\tau_n \leftarrow \ell_n \cdot t_n \cdot p(\mathbf{z}_n)/q(\mathbf{z}_n)$.            $\triangleright$ Note that we omit the normalization factor and $J$.
    **if** $\tau_n \leqslant \tau^*$ **then**
      $\tau^* \leftarrow \tau_n, j^* \leftarrow j_n, \tilde{n}^* \leftarrow \tilde{n}_{j_n}$.
    **end if**
    # check stopping criterion:
    **if** $t_n/r'_{\max} > \tau^*$ **then**
      **break**
    **end if**
  **end for**

---

## A.2 Practical ORC Algorithm with Space Partitioning

---

**Algorithm 4** Practical encoding procedure of ORC with Space Partitioning

---

**Input:** Target $Q$, shared coding distribution $P$, number of candidate samples $N$, and shared random states $S_j$ for each partition $j$.
**Output:** Bin index $j^*$, local sample index $\tilde{n}^*$.

  # initialize:
  $\tau^* \leftarrow \infty, t_0 \leftarrow 0, j^* \leftarrow 0, \tilde{n}^* \leftarrow 0$.
  Partition space in $J$ "bins".
  $\tilde{n}_j \leftarrow 0$ for $j = 1, 2, \cdots, J$.
  # run ORC:
  **for** $n = 1, 2, \cdots, N$ **do**
    # sample time:
    Sample $\Delta t_n \sim \text{Exp}(1)$.
    $t_n \leftarrow t_{n-1} + \frac{N}{N-n+1}\Delta t_n$.             $\triangleright$ Note the difference from the PFR algorithm.
    # simulate a sample with PRNG:
    $\mathbf{z} \sim Q$, and find $j_n$ s.t. $\mathbf{z} \in B_{j_n}$.
    $\tilde{n}_{j_n} \leftarrow \tilde{n}_{j_n} + 1$.
    $\mathbf{z}_n \leftarrow \text{PRNG}(P|_{B_{j_n}}, S_{j_n}, \tilde{n}_{j_n})$.
        $\triangleright$ by inverse transform sampling when both of the partitioning and $P$ are axis-aligned.
    # update $\tau^*$:
    $\tau_n \leftarrow J \cdot Q(B_{j_n}) \cdot t_n \cdot p(\mathbf{z}_n)/q(\mathbf{z}_n)$.
    **if** $\tau_n \leqslant \tau^*$ **then**
      $\tau^* \leftarrow \tau_n, j^* \leftarrow j_n, \tilde{n}^* \leftarrow \tilde{n}_{j_n}$.
    **end if**
  **end for**

---

## A.3 Decoding Algorithm

---

**Algorithm 5** Dncoding procedure of PFR/ORC with Space Partitioning

---

**Input:** Bin index $j^*$, local sample index $\tilde{n}^*$, and shared random states $S_j$ for each partition $j$.
**Output:** Sample $\mathbf{z}$.

  Partition space in $J$ "bins".
  $\mathbf{z} \leftarrow \text{PRNG}(P|_{B_{j*}}, S_{j*}, \tilde{n}^*)$.

---

# B Additional Results and Discussions

## B.1 Elucidating Non-overlapping Bins

As we discussed in the main text, the partitioning does not need to form non-overlapping grids. **In fact, there are only two constraints for the partitions: (1) the density of the distributions for all "bins", without being weighted by $\pi$, should sum to $p(\mathbf{z})$ at each z, and (2) integrating the density of z in each "bin" should yield the same constant $1/J$.** As long as we follow these two constraints, we can create any kind of partition, no matter whether they overlap or not. In Section 3.3, we explain this by viewing Equation (7) as a mixture model. In this section, we provide another explanation from an intuitive point of view.

First, we assume the reader agrees that we can always create non-overlapping bins as non-overlapped grids. Then, to create overlapping bins, we can add an auxiliary axis $x_{\text{aux}}$ to the original space $\Omega$, forming an augmented space as $x_{\text{aux}} \times \Omega$. We define the prior and target in the auxiliary axis as $P_{\text{aux}}$ and $Q_{\text{aux}}$, and define the prior and target in the augmented space as $P_{\text{aug}} = P_{\text{aux}} \times P$, $Q_{\text{aug}} = Q_{\text{aux}} \times Q$. This augmentation will not influence the codelength since we can always ensure the augmented $P_{\text{aug}}$ and $Q_{\text{aug}}$ to have identical marginal densities along the auxiliary axis. We illustrate this augmented

space in Figure 5a. We can view partitions in the original space as non-overlapping ones in an augmented space, as illustrated in Figures 5b to 5d.

In this augmented space, the partitions show follow these two constraints: (1) without considering $\pi$, if we marginalize our the auxiliary axis, $P_{\text{aug}}$ should revert to the original $P$; (2) we require all bins to have the same probability under the prior $P_{\text{aug}}$. The second constraint comes from the first step in Section 3.1, which is necessary for the proof of the codelength later in Equation (38). If we only consider the original space, these two constraints correspond to the ones aforementioned at the beginning of this section.

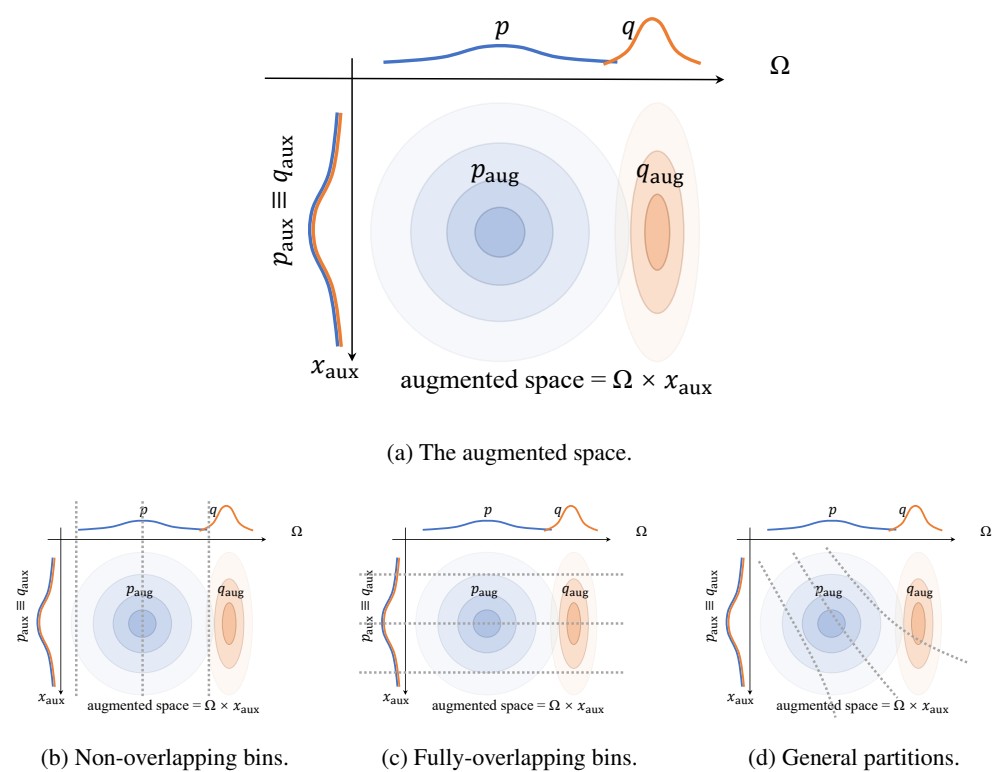

(a) The augmented space.

(b) Non-overlapping bins.  (c) Fully-overlapping bins.  (d) General partitions.

Figure 5: Elucidating the generality of space partitioning. $\Omega$ represents the original space. We use dashed lines to represent the boundaries of the partitions. (a) We can add an auxiliary axis $x_{\text{aux}}$ to the original space, forming an augmented space. We define the prior and target in the auxiliary axis as $P_{\text{aux}}$ and $Q_{\text{aux}}$, and define the prior and target in the augmented space as $P_{\text{aug}} = P_{\text{aux}} \times P$, $Q_{\text{aug}} = Q_{\text{aux}} \times Q$. We also require $P_{\text{aux}}$ and $Q_{\text{aux}}$ to be the same, so that $D_{\text{KL}}[Q_{\text{aug}} || P_{\text{aug}}]$ is the same as the original KL $D_{\text{KL}}[Q \parallel P]$. Dividing the augmented space into non-overlapping bins will lead to non-overlapping bins or overlapping bins in the original space. For example, as shown in (b), dividing the augmented space into non-overlapping bins whose boundaries are parallel to the auxiliary axis results in the standard non-overlapping bins in the original space $\Omega$. As shown in (c), dividing the augmented space into non-overlapping bins whose boundaries are orthogonal to the auxiliary axis results in fully overlapping bins in the original space $\Omega$. Also, as in (d), the augmented space can be divided in an arbitrary manner, leading to generally overlapping bins in the original space $\Omega$.

## B.2 Choosing the Number of Intervals Assigned to Each Axis

First, we elaborate on why this question matters. We take the following example: $Q$ and $P$ are 2D Gaussians, and they have the same marginal distribution along the first dimension, denoted as $x_1$, but have different marginal distributions along the second dimension, denoted as $x_2$.

Now, we partition the space $x_1 \times x_2$ by axis-aligned grids, i.e., the boundaries of grids are parallel to the axes. If we partition the space by solely dividing $x_1$, we will find that the prior and the posterior have the same probability for each bin, i.e., $P(B_j) = Q(B_j), \forall j$. In this case, choosing

$\pi(j) = Q(B_j)$ as discussed in Section 4.2, we can write the density of $P'$ (Equation (7)) as

$$p'(\mathbf{z}) = \sum_{j=1}^{J} \mathbb{1}\{\mathbf{z} \in B_j\} \frac{\pi(j)p(\mathbf{z})}{P(B_j)} \tag{16}$$

$$= \sum_{j=1}^{J} \mathbb{1}\{\mathbf{z} \in B_j\} \frac{Q(j)p(\mathbf{z})}{P(B_j)} \tag{17}$$

$$= \sum_{j=1}^{J} \mathbb{1}\{\mathbf{z} \in B_j\} \frac{\cancel{P(B_j)}p(\mathbf{z})}{\cancel{P(B_j)}} \tag{18}$$

$$= \sum_{j=1}^{J} \mathbb{1}\{\mathbf{z} \in B_j\}p(\mathbf{z}) \tag{19}$$

$$= p(\mathbf{z}) \tag{20}$$

which means that the adjusted search heuristic $P'$ is the same as the original prior $P$, and thus we will have no improvements in the runtime. Figure 6a visualize this example.

On the other hand, if we partition the space by dividing $x_2$, as shown in Figure 6b, the search heuristic $P'$ will align with $Q$ better than $P$, and hence we can reduce the runtime. In practice, it is not always feasible to partition in this way. Instead, we may partition the space by dividing both $x_1$ and $x_2$, as shown in Figure 6c. In this case, we will still have reduced runtime. This is because

$$D_{\mathrm{KL}}[Q \parallel P'] = D_{\mathrm{KL}}[Q \parallel P] - \sum_{j=1}^{J} Q(B_j) \log_2 \pi(j) + \sum_{j=1}^{J} Q(B_j) \log_2 P(B_j) \tag{21}$$

$$= D_{\mathrm{KL}}[Q \parallel P] - \left( \sum_{j=1}^{J} Q(B_j) \log_2 Q(j) - \sum_{j=1}^{J} Q(B_j) \log_2 P(B_j) \right) \tag{22}$$

We recognize the term $\left( \sum_{j=1}^{J} Q(B_j) \log_2 Q(j) - \sum_{j=1}^{J} Q(B_j) \log_2 P(B_j) \right)$ as a KL divergence and hence will be non-negative. Whene there exists $B_j$, s.t. $Q(B_j) \neq P(B_j)$, the KL term $\left( \sum_{j=1}^{J} Q(B_j) \log_2 Q(j) - \sum_{j=1}^{J} Q(B_j) \log_2 P(B_j) \right)$ will always be positive to ensure $D_{\mathrm{KL}}[Q \parallel P'] \leqslant D_{\mathrm{KL}}[Q \parallel P]$ and hence guarantee a reduction in the runtime.

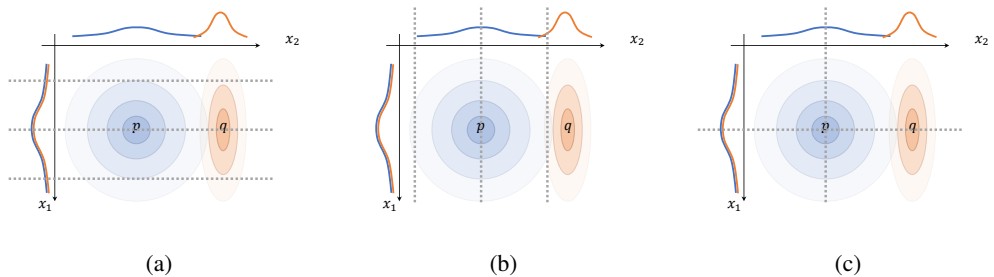

|         (a)          |         (b)          |         (c)          |

Figure 6: An example to explain why the number of intervals assigned to each axis matters when we partition the space with axis-aliged grids. We use dashed lines to represent the boundaries of the partitions. In this example, $Q$ and $P$ have the same marginal along $x_1$. In (a), we partition the space by solely dividing $x_1$. This will make the search heuristic $P'$ equal to $P$, leading to no improvements in the runtime. In (b), we partition the space by dividing $x_2$. This will make the search heuristic $P'$ align with $Q$ better than $P$ (i.e., $D_{\mathrm{KL}[Q||P']} < D_{\mathrm{KL}[Q||P]}$), reducing the runtime. In (c), we partition the space by dividing both $x_2$ and $x_1$. This will still reduce the runtime, but is not as efficient as (b). Please note that while the plot looks similar to Figure 5, they represent different concepts.

From this example, we conclude that even when the total number of partitions is fixed, different partition strategies will lead to different runtime. What we want to avoid is the case in Figure 6a. Fortunately, in most neural compression applications, we have access to an estimation of the mutual information $I_d$ along each axis $d$ from the training set. For the reader who is not familiar with the term, we can view $I_d$ as the average KL divergence along $d$-th axis. If the mutual information is *non-zero* along one axis, on average, $Q$ and $P$ will *not* have the same marginal distribution along this axis. Therefore, we can partition the $d$-th axis into approximately $2^{\wedge}\left( I_d \cdot D_{\mathrm{KL}}[Q \parallel P] / \sum_{d'=1}^{D} I_{d'} \right)$ intervals to avoid the worst case in Figure 6a.

### B.2.1 Examples and Ablations on Choosing the Number of Intervals Assigned to Each Axis

**Qualitative visualization**. We first visualize the approximation error of using ORC with different partitioning strategies on a toy problem. Specifically, we create a 5D Gaussian target, and we set dimension 1 to have 0 mutual information (we call it a collapsed/uninformative dimension). We compare three partition strategies: only partitioning the collapsed dimension, randomly assigning intervals to dimensions, and assigning intervals according to mutual information (our proposed strategy). We also include standard ORC for reference. We run the four settings with the same number of samples (20) and repeat each setting 5000 times. We show the histogram of 5000 encoded results in Figure 7. As we can see, assigning intervals according to mutual information works best, whereas partitioning only the collapsed dimension yields almost the same results as standard ORC. This verifies our discussion above.

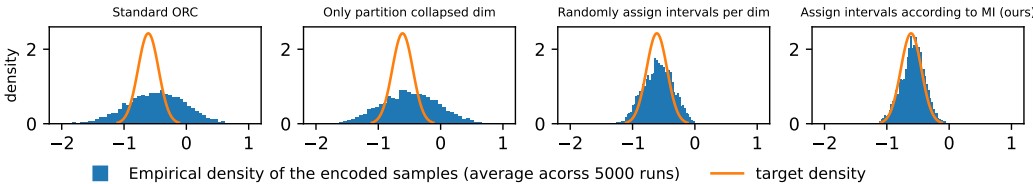

Figure 7: Visualizing approximation error of standard ORC and our proposed methods with different partitioning strategies when executed with the same number of samples (20). We use the same setup as the toy experiments in the main text (details in Appendix D.1), but here we set dimension 1 to have zero mutual information (i.e., collapsed dimension).

**Ablation study**. We now provide ablation studies showing how partition strategies will influence performance. We run the ablation on the Gaussian toy example and the CIFAR-10 compression experiments. As we can see, our proposed partition strategy is always better than randomly assigning intervals per dim. For CIFAR-10, we also compare the results by first removing uninformative dimensions (mutual information 0), and then randomly assigning intervals to other dimensions. We find this is only slightly worse than our proposed partition strategy. This not only further verifies our discussion above in Appendix B.2, but also indicates that our algorithm is not very sensitive to how we construct the intervals as long as we avoid partitioning along uninformative dimensions.

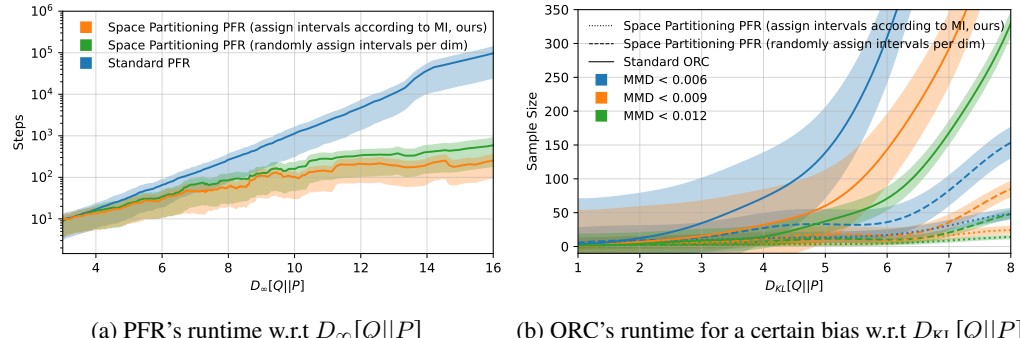

(a) PFR's runtime w.r.t $D_\infty[Q||P]$      (b) ORC's runtime for a certain bias w.r.t $D_{\mathrm{KL}}[Q||P]$

Figure 8: Runtime (number of steps/simulated samples) of different partition strategies on 5D toy Gaussian examples. We compare two partition strategies: randomly assigning intervals to dimensions and assigning intervals according to mutual information. We also include standard PFR and ORC for reference.

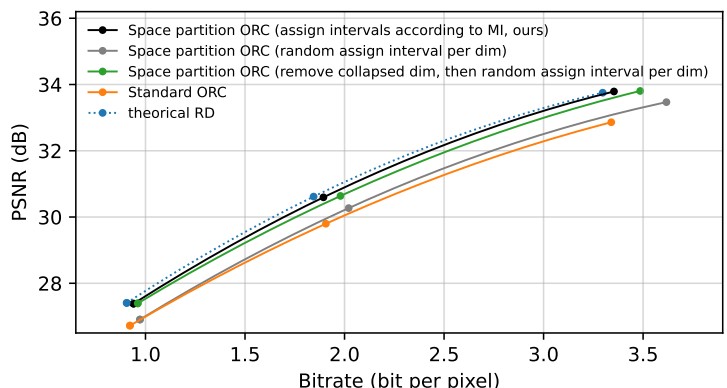

Figure 9: Rate-distortion of RECOMBINER (on 100 CIFAR-10 test images) by our space-partitioning algorithm using different partition strategies: randomly assigning intervals to dimensions; first removing axis with MI $\approx 0$ and then randomly assigning intervals; assigning intervals according to MI (our proposed strategy). We include standard ORC and theoretical RD for reference. Our proposed partition strategy is better than randomly assigning intervals per axis. Surprisingly, the results achieved by first removing uninformative dimensions (MI $\approx 0$) and then randomly assigning intervals to other dimensions are only slightly worse than our proposed partition strategy. This indicates that our algorithm is not very sensitive to how we construct the intervals, as long as we avoid partitioning along uninformative dimensions.

## C  Proofs and Derivations

### C.1  Proof of Theorem 3.1

Restate Theorem 3.1 for easy reference:

**Theorem 3.1.** *Let a pair of correlated random variables* $\mathbf{X}, \mathbf{Z} \sim P_{\mathbf{X}, \mathbf{Z}}$ *be given. Assume we perform relative entropy coding using Algorithm 2 and let* $j^*$ *denote the bin index and* $\tilde{n}^*$ *the local sample index returned by the algorithm. Then, the entropy of the two-part code is bounded by:*

$$\mathbb{H}[j^*, \tilde{n}^*] \leqslant I[\mathbf{X}; \mathbf{Z}] + \mathbb{E}_{\mathbf{X}}[\epsilon] + \log_2(I[\mathbf{X}; \mathbf{Z}] - \log_2 J + \mathbb{E}_{\mathbf{X}}[\epsilon] + 1) + 4, \tag{23}$$

$$\text{where} \quad \epsilon = \mathbb{E}_{\mathbf{z} \sim Q_{\mathbf{Z}|\mathbf{X}}} \left[ \max \left\{ 0, \log_2 J - \log_2 \frac{q(\mathbf{z})}{p(\mathbf{z})} \right\} \right]. \tag{24}$$

*Proof.* The proof will be organized as follows: first, we will prove an upper bound of $\mathbb{E}[\tilde{n}^*|\mathbf{X} = \mathbf{x}]$ for the one-shot case, i.e., the expectation of $\tilde{n}^*$ given a realization of the data $\mathbf{X}$. Then, we will take the expectation over $\mathbf{X}$ to achieve an upper bound for the average case $\mathbb{E}[\tilde{n}^*]$. Finally, we will consider encoding $\tilde{n}^*$ using a Zipf distribution to calculate the upper bound given in Theorem 3.1.

**Derive** $\mathbb{E}[\tilde{n}^*|\mathbf{X} = \mathbf{x}]$**.** Assume that the posterior distribution $\mathbf{Z}|\mathbf{X} = \mathbf{x}$ is $Q$. We run PFR with space partitioning according to Algorithm 2 using a coding distribution $P$ to encode a sample from $Q$. As we discussed in the main text, we can view our proposed algorithm as running standard PFR with an adjusted prior $P'$, whose density is defined as

$$p'(\mathbf{z}) = \sum_{j=1}^{J} \mathbb{1}\{\mathbf{z} \in B_j\} \frac{\pi(j)p(\mathbf{z})}{P(B_j)}. \tag{25}$$

Therefore, we follow similar arguments as Li and El Gamal (2018) to prove the codelength.

We first assume the minimal $\tau$ is $\tau^*$, the corresponding sample is $\mathbf{z}^*$, and the global sample index is $n^*$. WLG, let's assume the sample falls in bin $j^*$ and its local sample index in this bin is $\tilde{n}^*$.

According to Li and El Gamal (2018), we have

$$\tau^* \sim \mathrm{Exp}(1) \tag{26}$$

$$n^* - 1 \sim \mathrm{Poisson}(\lambda) \tag{27}$$

$$\lambda \leqslant \tau^* \frac{q(\mathbf{z}^*)}{p'(\mathbf{z}^*)} \tag{28}$$

Equation (28) is obtained from Appendix A in Li and El Gamal (2018) and replacing $p$ by $p'$.

Additionally, we have

$$\tilde{n}^* - 1 | n^* - 1 \sim \mathrm{Binomial}(\tilde{n}^* - 1 | n^* - 1, \pi(j^*)). \tag{29}$$

This is because: if we define "success" as a sample falling in the bin $j^*$, then the probability that the sample in the bin $j^*$ with global sample index $n^*$ has a local sample index $\tilde{n}^*$ is equivalent to the probability that there are $\tilde{n}^* - 1$ success in the first $n^* - 1$ Bernoulli trials.

Therefore, according to the property of Poisson distribution, we have

$$\tilde{n}^* - 1 \sim \mathrm{Poisson}(\pi(j^*)\lambda) \tag{30}$$

Therefore, we have

$$\mathbb{E}\log_2[\tilde{n}^*|\mathbf{X} = \mathbf{x}] = \mathbb{E}_{j^*}\mathbb{E}_{\mathbf{z}^* \sim Q|_{B_{j^*}}}\mathbb{E}_{\tau^*}\mathbb{E}_{\tilde{n}^*|j^*,\mathbf{z},\tau^*}\left[\log_2(\tilde{n}^* - 1 + 1)\right] \tag{31}$$

$$\overset{\text{Jensen's}}{\leqslant} \mathbb{E}_{j^*}\mathbb{E}_{\mathbf{z}^* \sim Q|_{B_{j^*}}}\mathbb{E}_{\tau^*}\left[\log_2\mathbb{E}_{\tilde{n}^*|j^*,\mathbf{z},\tau^*}(\tilde{n}^* - 1 + 1)\right] \tag{32}$$

$$\overset{\text{eq. (30)}}{\leqslant} \mathbb{E}_{j^*}\mathbb{E}_{\mathbf{z}^* \sim Q|_{B_{j^*}}}\mathbb{E}_{\tau^*}\left[\log_2(\pi(j^*)\lambda + 1)\right] \tag{33}$$

$$\overset{\text{eq. (28)}}{\leqslant} \mathbb{E}_{j^*}\mathbb{E}_{\mathbf{z}^* \sim Q|_{B_{j^*}}}\mathbb{E}_{\tau^*}\log_2\left[\tau^*\pi(j^*)\frac{q(\mathbf{z}^*)}{p'(\mathbf{z}^*)} + 1\right] \tag{34}$$

$$\overset{\text{Jensen's}}{\leqslant} \mathbb{E}_{j^*}\mathbb{E}_{\mathbf{z}^* \sim Q|_{B_{j^*}}}\log_2\left[\pi(j^*)\frac{q(\mathbf{z}^*)}{p'(\mathbf{z}^*)} + 1\right] \tag{35}$$

$$\overset{\text{eq. (25)}}{=} \mathbb{E}_{j^*}\mathbb{E}_{\mathbf{z}^* \sim Q|_{B_{j^*}}}\log_2\left[\cancel{\pi(j^*)}\frac{q(\mathbf{z}^*)P(B_{j^*})}{p(\mathbf{z}^*)\cancel{\pi(j^*)}} + 1\right] \tag{36}$$

$$= \mathbb{E}_{\mathbf{z}^* \sim Q}\log_2\left[\frac{q(\mathbf{z}^*)P(B_{j^*})}{p(\mathbf{z}^*)} + 1\right] \tag{37}$$

$$= \mathbb{E}_{\mathbf{z} \sim Q}\log_2\left[\frac{q(\mathbf{z})}{Jp(\mathbf{z})} + 1\right] \tag{38}$$

The last line is because we require $P(B_1) = P(B_2) = \cdots = P(B_J) = 1/J$. We can see that Equation (38) does not depend on either the specific space partitioning strategy or the choice of $\pi$.

Based on Taylor expansion of $\log_2(1 + x)$ at $0, +\infty$, we have

$$\log_2\left(1 + \frac{x}{a}\right) \leqslant \begin{cases} \frac{x}{a}\log_2 e, & 0 \leqslant x \leqslant a \\ \log_2\left(\frac{x}{a}\right) + \frac{a}{x}\log_2 e, & \text{otherwise} \end{cases} \tag{39}$$

Therefore,

$$\mathbb{E}\log_2[\tilde{n}^* | \mathbf{X} = \mathbf{x}] \leqslant \mathbb{E}_{\mathbf{z} \sim Q} \log_2 \left[ \frac{q(\mathbf{z})}{J p(\mathbf{z})} + 1 \right] \tag{40}$$

$$\leqslant \mathbb{E}_{\mathbf{z} \sim Q} \log_2 \left[ \mathbb{1} \left\{ \frac{q(\mathbf{z})}{p(\mathbf{z})} \leqslant J \right\} \left( \frac{q(\mathbf{z})}{J p(\mathbf{z})} \right) \log_2 e \right.$$
$$\left. + \mathbb{E}_{\mathbf{z} \sim Q} \log_2 \left[ \mathbb{1} \left\{ \frac{q(\mathbf{z})}{p(\mathbf{z})} > J \right\} \left( \log_2 \frac{q(\mathbf{z})}{J p(\mathbf{z})} + \frac{J p(\mathbf{z})}{q(\mathbf{z})} \log_2 e \right) \right] \tag{41}$$

$$= \mathbb{E}_{\mathbf{z} \sim Q} \log_2 \left[ \mathbb{1} \left\{ \frac{q(\mathbf{z})}{p(\mathbf{z})} > J \right\} \left( \log_2 \frac{q(\mathbf{z})}{J p(\mathbf{z})} \right) \right]$$
$$+ \log_2 e \cdot \mathbb{E}_{\mathbf{z} \sim Q} \log_2 \left[ \mathbb{1} \left\{ \frac{q(\mathbf{z})}{p(\mathbf{z})} \leqslant J \right\} \left( \frac{q(\mathbf{z})}{J p(\mathbf{z})} \right) + \mathbb{1} \left\{ \frac{q(\mathbf{z})}{p(\mathbf{z})} > J \right\} \left( \frac{J p(\mathbf{z})}{q(\mathbf{z})} \right) \right] \tag{42}$$

$$\leqslant \mathbb{E}_{\mathbf{z} \sim Q} \log_2 \left[ \mathbb{1} \left\{ \frac{q(\mathbf{z})}{p(\mathbf{z})} > J \right\} \left( \log_2 \frac{q(\mathbf{z})}{J p(\mathbf{z})} \right) \right] + \log_2 e \tag{43}$$

The last line is because

$$\mathbb{E}_{\mathbf{z} \sim Q} \log_2 \left[ \mathbb{1} \left\{ \frac{q(\mathbf{z})}{p(\mathbf{z})} \leqslant J \right\} \left( \frac{q(\mathbf{z})}{J p(\mathbf{z})} \right) + \mathbb{1} \left\{ \frac{q(\mathbf{z})}{p(\mathbf{z})} > J \right\} \left( \frac{J p(\mathbf{z})}{q(\mathbf{z})} \right) \right] \tag{44}$$

$$\leqslant \mathbb{E}_{\mathbf{z} \sim Q} \log_2 \left[ \mathbb{1} \left\{ \frac{q(\mathbf{z})}{p(\mathbf{z})} \leqslant J \right\} \cdot 1 + \mathbb{1} \left\{ \frac{q(\mathbf{z})}{p(\mathbf{z})} > J \right\} \cdot 1 \right] \tag{45}$$

$$= 1 \tag{46}$$

We can further simplify Equation (43) as follows:

$$\mathbb{E}\log_2[\tilde{n}^* | \mathbf{X} = \mathbf{x}] \leqslant D_{\mathrm{KL}}[Q || P] - \log_2 J + \mathbb{E}_Q \left[ \mathbb{1} \left\{ \log_2 \frac{q(\mathbf{z})}{p(\mathbf{z})} \leqslant \log_2 J \right\} \left( \log_2 J - \log_2 \frac{q(\mathbf{z})}{p(\mathbf{z})} \right) \right] + \log_2 e \tag{47}$$

$$= D_{\mathrm{KL}}[Q || P] - \log_2 J + \epsilon + \log_2 e \tag{48}$$

where we denote

$$\epsilon = \mathbb{E}_{\mathbf{z} \sim Q} \left[ \mathbb{1} \left\{ \log_2 \frac{q(\mathbf{z})}{p(\mathbf{z})} \leqslant \log_2 J \right\} \left( \log_2 J - \log_2 \frac{q(\mathbf{z})}{p(\mathbf{z})} \right) \right] \tag{49}$$

$$= \mathbb{E}_{\mathbf{z} \sim Q} \left[ \max \left\{ \log_2 J - \log_2 \frac{q(\mathbf{z})}{p(\mathbf{z})}, 0 \right\} \right] \tag{50}$$

**Derive $\mathbb{E}[\tilde{n}^*]$.** Now, we consider the average case by taking the expectation over $\mathbf{X}$. We have

$$\mathbb{E}[\tilde{n}^*] \leqslant \mathbb{E}_{\mathbf{X}} D_{\mathrm{KL}}[Q_{\mathbf{Z}|\mathbf{X}} || P_{\mathbf{Z}}] - \log_2 J + \mathbb{E}_{\mathbf{X}}[\epsilon] + \log_2 e \tag{51}$$

$$= I[\mathbf{X}; \mathbf{Z}] - \log_2 J + \mathbb{E}_{\mathbf{X}}[\epsilon] + \log_2 e \tag{52}$$

**Derive entropy.** Following the maximum entropy argument similar to Li and El Gamal (2018), we have

$$\mathbb{H}[\tilde{n}^*] \leqslant I[\mathbf{X}; \mathbf{Z}] - \log_2 J + \mathbb{E}_{\mathbf{X}}[\epsilon] + \log_2 e + \log_2(I[\mathbf{X}; \mathbf{Z}] - \log_2 J + \mathbb{E}_{\mathbf{X}}[\epsilon] + \log_2 e + 1) + 1 \tag{53}$$

$$\leqslant I[\mathbf{X}; \mathbf{Z}] - \log_2 J + \mathbb{E}_{\mathbf{X}}[\epsilon] + \log_2(I[\mathbf{X}; \mathbf{Z}] - \log_2 J + \mathbb{E}_{\mathbf{X}}[\epsilon] + 1) + 1 + \log_2 e + \log_2(\log_2 e + 1) \tag{54}$$

$$\leqslant I[\mathbf{X}; \mathbf{Z}] - \log_2 J + \mathbb{E}_{\mathbf{X}}[\epsilon] + \log_2(I[\mathbf{X}; \mathbf{Z}] - \log_2 J + \mathbb{E}_{\mathbf{X}}[\epsilon] + 1) + 4 \tag{55}$$

Finally, we need $\log_2 J$ bits to encode the index $j^*$. Therefore, the two-part code's codelength is upper-bounded by

$$\mathbb{H}[\tilde{n}^*] + \log_2 J \leqslant I[\mathbf{X}; \mathbf{Z}] + \mathbb{E}_{\mathbf{X}}[\epsilon] + \log_2(I[\mathbf{X}; \mathbf{Z}] - \log_2 J + \mathbb{E}_{\mathbf{X}}[\epsilon] + 1) + 4 \tag{56}$$

$$\square$$

We also note that, if we do not take the expectation over $\mathbf{X}$, we can also achieve a similar bound for the one-shot channel simulation task targeting $Q$ using a coding distribution $P$:

$$\mathbb{H}[\tilde{n}^*] + \log_2 J \leqslant D_{\mathrm{KL}}[Q||P] + \epsilon + \log_2(D_{\mathrm{KL}}[Q||P] - \log_2 J + \epsilon + 1) + 4. \tag{57}$$

This ensures that the codelength is consistent for each realization of $\mathbf{X}$ and guarantees a good encoding performance both on average and for each certain data observation in practice.

## C.2 Proof of Proposition 3.2

To prove Proposition 3.2, we first prove the following Lemma:

**Lemma C.1.** *The $\epsilon$-cost defined in Equation* (9) *monotonically increases w.r.t $J$.*

*Proof.* By setting $r = \log_2 {}^{q(\mathbf{z})}/_{p(\mathbf{z})}$ which is a new random variable, we rewrite the $\epsilon$-cost as

$$\mathbb{E}_{\mathbf{z}\sim Q}\left[\mathbb{1}\left\{\log_2 \frac{q(\mathbf{z})}{p(\mathbf{z})} \leqslant \log_2 J\right\}\left(\log_2 J - \log_2 \frac{q(x)}{p(x)}\right)\right] \tag{58}$$

$$= \mathbb{E}_r\left[\mathbb{1}\left\{r \leqslant \mathcal{J}\right\}(\mathcal{J} - r)\right] \tag{59}$$

$$= \int_{-\infty}^{\mathcal{J}} p(r)(\mathcal{J} - r)dr \tag{60}$$

where we also write $\mathcal{J} = \log_2 J$, which is a monotonically increasing function of $J$. Take derivative w.r.t. $\mathcal{J}$, we have

$$d\frac{\int_{-\infty}^{\mathcal{J}} p(r)(\mathcal{J} - r)dr}{d\mathcal{J}} = p(\mathcal{J})(\mathcal{J} - \mathcal{J}) + \int_{-\infty}^{\mathcal{J}} p(r)dr = \int_{-\infty}^{\mathcal{J}} p(r)dr \geqslant 0 \tag{61}$$

which finishes the proof. $\qquad\square$

Therefore, to prove Proposition 3.2, we only need to look at the worst case, where $J = 2^{D_{\mathrm{KL}}[Q||P]}$. We now prove for Uniform and Gaussian, respectively:

**Uniform.** For uniform distribution, WLG, assume $P = \mathcal{U}(0,1)^n$, and $Q = \mathcal{U}(A)$, where $A \subseteq (0,1)^n$. In this case, we have $D_{\mathrm{KL}}[Q||P] = -\log_2 \mu(A)$, and $\frac{q(\mathbf{z})}{p(\mathbf{z})} = \frac{1}{\mu(A)}, \forall \mathbf{z} \in A$, where $\mu$ is the Lesbague measure. Therefore,

$$\epsilon = \mathbb{E}_{\mathbf{z}\sim Q}\left[\mathbb{1}\left\{\log_2 \frac{q(\mathbf{z})}{p(\mathbf{z})} \leqslant \log_2 J\right\}\left(\log_2 J - \log_2 \frac{q(\mathbf{z})}{p(\mathbf{z})}\right)\right] \tag{62}$$

$$\leqslant \mathbb{E}_{\mathbf{z}\sim Q}\left[\mathbb{1}\left\{\log_2 \frac{q(\mathbf{z})}{p(\mathbf{z})} \leqslant D_{\mathrm{KL}}[Q||P]\right\}\left(D_{\mathrm{KL}}[Q||P] - \log_2 \frac{q(\mathbf{z})}{p(\mathbf{z})}\right)\right] \tag{63}$$

$$= 0 \tag{64}$$

On the other hand, the indicator function ensures the integrand is non-negative, and hence $\epsilon \geqslant 0$. Therefore, $\epsilon = 0$.

**Factorized Gaussian.** To prove for Gaussian, we first prove the following lemma:

**Lemma C.2.** *When $J = 2^{D_{KL}[Q||P]}$, the $\epsilon$-cost defined in Equation* (9) *is upper-bounded by*

$$\epsilon \leqslant \frac{1}{2}\sqrt{\mathrm{Var}[r]}, \tag{65}$$

*where $r$ is the RV defined by $r = \log_2 {}^{q(\mathbf{z})}/_{p(\mathbf{z})}$, $\mathbf{z} \sim Q$.*

*Proof.* First, we know

$$\mathbb{E}_{\mathbf{z}\sim Q}\left[D_{\mathrm{KL}}[Q||P] - \log_2 \frac{q(\mathbf{z})}{p(\mathbf{z})}\right] = 0 \tag{66}$$

and we recognize

$$\mathbb{E}_{\mathbf{z}\sim Q}\left[D_{\mathrm{KL}}[Q\|P] - \log_2 \frac{q(\mathbf{z})}{p(\mathbf{z})}\right] \tag{67}$$

$$=\mathbb{E}_{\mathbf{z}\sim Q}\left[\mathbb{1}\left\{\log_2 \frac{q(\mathbf{z})}{p(\mathbf{z})} \leqslant D_{\mathrm{KL}}[Q\|P]\right\}\left(D_{\mathrm{KL}}[Q\|P] - \log_2 \frac{q(\mathbf{z})}{p(\mathbf{z})}\right)\right]$$

$$+ \mathbb{E}_{\mathbf{z}\sim Q}\left[\mathbb{1}\left\{\log_2 \frac{q(\mathbf{z})}{p(\mathbf{z})} > D_{\mathrm{KL}}[Q\|P]\right\}\left(D_{\mathrm{KL}}[Q\|P] - \log_2 \frac{q(\mathbf{z})}{p(\mathbf{z})}\right)\right] \tag{68}$$

When $J = 2^{D_{\mathrm{KL}}[Q\|P]}$, we have

$$\epsilon =\mathbb{E}_{\mathbf{z}\sim Q}\left[\max\left\{0, D_{\mathrm{KL}}[Q\|P] - \log_2 \frac{q(\mathbf{z})}{p(\mathbf{z})}\right\}\right] \tag{69}$$

$$\mathbb{E}_{\mathbf{z}\sim Q}\left[\mathbb{1}\left\{\log_2 \frac{q(\mathbf{z})}{p(\mathbf{z})} \leqslant D_{\mathrm{KL}}[Q\|P]\right\}\left(D_{\mathrm{KL}}[Q\|P] - \log_2 \frac{q(\mathbf{z})}{p(\mathbf{z})}\right)\right] \tag{70}$$

$$=\mathbb{E}_{\mathbf{z}\sim Q}\left[\mathbb{1}\left\{\log_2 \frac{q(\mathbf{z})}{p(\mathbf{z})} > D_{\mathrm{KL}}[Q\|P]\right\}\left(\log_2 \frac{q(\mathbf{z})}{p(\mathbf{z})} - D_{\mathrm{KL}}[Q\|P]\right)\right] \tag{71}$$

Therefore, we have

$$2 \cdot \epsilon = \mathbb{E}_{\mathbf{z}\sim Q}\left[\mathbb{1}\left\{\log_2 \frac{q(\mathbf{z})}{p(\mathbf{z})} \leqslant D_{\mathrm{KL}}[Q\|P]\right\}\left(D_{\mathrm{KL}}[Q\|P] - \log_2 \frac{q(\mathbf{z})}{p(\mathbf{z})}\right)\right]$$

$$+ \mathbb{E}_{\mathbf{z}\sim Q}\left[\mathbb{1}\left\{\log_2 \frac{q(\mathbf{z})}{p(\mathbf{z})} > D_{\mathrm{KL}}[Q\|P]\right\}\left(\log_2 \frac{q(\mathbf{z})}{p(\mathbf{z})} - D_{\mathrm{KL}}[Q\|P]\right)\right] \tag{72}$$

$$= \mathbb{E}_{\mathbf{z}\sim Q}\left[\left|\log_2 \frac{q(\mathbf{z})}{p(\mathbf{z})} - D_{\mathrm{KL}}[Q\|P]\right|\right] \tag{73}$$

Writing $r = \log_2 q(\mathbf{z})/p(\mathbf{z})$, we have

$$\epsilon = \frac{1}{2}\mathbb{E}_r\left\{|r - \mathbb{E}[r]|\right\} \tag{74}$$

$$= \frac{1}{2}\mathbb{E}_r\left\{\sqrt{(r - \mathbb{E}[r])^2}\right\} \tag{75}$$

$$\overset{\text{Jensen's}}{\leqslant} \frac{1}{2}\sqrt{\mathbb{E}_r\left\{(r - \mathbb{E}[r])^2\right\}} \tag{76}$$

$$= \frac{1}{2}\sqrt{\mathrm{Var}[r]} \tag{77}$$

$$\square$$

Lemma C.2 means that the $\epsilon$-cost is closely related to the variance of the log-density ratio between the target and the prior. *We highlight that this conclusion holds generally and can be used to prove the bound of $\epsilon$ for all distribution families.* But for now, we only focus on Gaussian, and leave more extensions to future exploration.

According to Lemma C.2, if we want to bound $\epsilon$, we only need to bound $\mathrm{Var}[r]$. To achieve this, we state the following Lemma:

**Lemma C.3.** *For 1D Gaussian Q and P, with density $q = \mathcal{N}(\mu_q, \sigma_q^2)$, $p = \mathcal{N}(\mu_p, \sigma_p^2)$, and $\sigma_q \leqslant \sigma_p$, defining random variable $r$ by $r = \log_2 q(x)/p(x)$, $x \sim Q$, if $D_{KL}[Q\|P]$ is fixed to a constant, then $\mathrm{Var}[r]$ monotonically increases w.r.t $\sigma_q$.*

*Proof.* WLG, let's assume $p = \mathcal{N}(0, 1)$, $\mu_q > 0$, $D_{\mathrm{KL}}[Q\|P] = \delta$. Denoting $v = \sigma_q^2$, we have

$$\mu_q = \sqrt{2\delta + \log(\sigma_q^2) - \sigma_q^2 + 1} = \sqrt{2\delta + \log(v) - v + 1} \tag{78}$$

and thus

$$r = \frac{1}{2v}\left((1 - v)x^2 - 2\mu_q x + \mu_q^2 - v + v\mu_q^2 + v^2\right) \tag{79}$$

$$= \frac{1}{2v}\left((1 - v)(x - \frac{\mu_q}{1 - v})^2 - \frac{\mu_q^2}{1 - v} + \mu_q^2 - v + v\mu_q^2 + v^2\right), \quad x \sim \mathcal{N}(\mu_q, v) \tag{80}$$

Reparameterizating $y = (x - \frac{\mu_q}{1-v})/\sqrt{v}$, we have

$$r = \frac{1}{2v}\left(v(1-v)y^2 - \frac{\mu_q^2}{1-v} + \mu_q^2 - v + v\mu_q^2 + v^2\right), y \sim \mathcal{N}(\frac{\mu_q}{\sqrt{v}} - \frac{\mu_q}{\sqrt{v}(1-v)}, 1) \quad (81)$$

The variance of $r$ is given by

$$\mathrm{Var}[r] = \frac{(1-v)^2}{4}\left[4\left(\frac{\mu_q}{\sqrt{v}} - \frac{\mu_q}{\sqrt{v}(1-v)}\right)^2 + 2\right] \quad (82)$$

$$= (1-v)^2\mu_q^2\left[\left(\frac{\sqrt{v}}{1-v}\right)^2 + \frac{1}{2}\right] \quad (83)$$

$$= v\mu_q^2 + \frac{1}{2}(1-v)^2\mu_q^2 \quad (84)$$

$$= \frac{1}{2}(v^2 + 1)(2\delta + \log(v) - v + 1) \quad (85)$$

Taking derivative, we have

$$\frac{d\mathrm{Var}[r]}{dv} = v(2\delta + \log(v) - v + 1) + \frac{1}{2}(v^2 + 1)(\frac{1}{v} - 1) \quad (86)$$

$$= v(2\delta + \log(v) - v + 1) + \frac{1}{2}(v + \frac{1}{v} - v^2 - 1) \quad (87)$$

$$\geqslant v(2\delta + 2 - \frac{1}{v} - v) + \frac{1}{2}(v + \frac{1}{v} - v^2 - 1) \quad (88)$$

$$= 2v\delta + \frac{5}{2}v + \frac{1}{2v} - \frac{3}{2}v^2 - \frac{3}{2} \quad (89)$$

$$\geqslant \frac{5}{2}v + \frac{1}{2v} - \frac{3}{2}v^2 - \frac{3}{2} \quad (90)$$

$$\geqslant 0(\text{when } v \leqslant 1) \quad (91)$$

$\square$

In summary, Lemma C.2 tells us if we want to upper-bound $\epsilon$, we only need to upper-bound $\mathrm{Var}[r]$; while Lemma C.3 further tells us if we want to upper-bound $\mathrm{Var}[r]$, we only need to look at the worst case, when $Q$ and $P$ share the same variance.

Following this, we can finally finish our prove:

WLG, assume the density of $Q$ and $P$ are given by

$$p(\mathbf{z}) = \mathcal{N}(\mathbf{z}|0, \mathbf{I}) \quad (92)$$
$$q(\mathbf{z}) = \mathcal{N}(z_1|\mu_1, 1)\mathcal{N}(z_2|\mu_2, 1)\cdots\mathcal{N}(z_d|\mu_d, 1) \quad (93)$$

where $d$ is the dimensionality. In this case, we have

$$D_{\mathrm{KL}}[Q||P] = \log_2 e \sum_{i=1}^n \frac{\mu_i^2}{2} \quad (94)$$

and

$$r = \log_2\frac{q(\mathbf{z})}{p(\mathbf{z})} = \log_2 e\left(\sum_{i=1}^n \mu_i z_i - \sum_{i=1}^n \frac{\mu_i^2}{2}\right) \sim \mathcal{N}\left(\log_2 e\sum_{i=1}^n \frac{\mu_i^2}{2}, \log_2^2 e\sum_{i=1}^n \mu_i^2\right) \quad (95)$$

Therefore,

$$\epsilon \leqslant \frac{1}{2}\sqrt{\mathrm{Var}[r]} = \sqrt{\frac{1}{4}\log_2^2 e\sum_{i=1}^n \mu_i^2} = \sqrt{\frac{\log_2 e}{2}D_{\mathrm{KL}}[Q||P]} \approx 0.849\sqrt{D_{\mathrm{KL}}[Q||P]} \quad (96)$$

Finally, taking expectation over $\mathbf{X}$, and by Jensen's Inequality, we have

$$\mathbb{E}_{\mathbf{X}}[\epsilon] \leqslant 0.849\mathbb{E}_{\mathbf{X}}\left[\sqrt{D_{\mathrm{KL}}[Q||P]}\right] \tag{97}$$

$$\leqslant 0.849\sqrt{\mathbb{E}_{\mathbf{X}}\left[D_{\mathrm{KL}}[Q||P]\right]} \tag{98}$$

$$= 0.849\sqrt{I[\mathbf{X};\mathbf{Z}]}, \tag{99}$$

which finishes the proof for the Gaussian case in Proposition 3.2.

## C.3    Proof of Theorem 3.3

We first restate Theorem 3.3 for easy reference:

**Theorem 3.3.** *Let a pair of correlated random variables $\mathbf{X}, \mathbf{Z} \sim P_{\mathbf{X},\mathbf{Z}}$ be given. Assume we perform relative entropy coding using GPRS with space partitioning and let $j^*$ denote the bin index and $\tilde{n}^*$ the local sample index returned by the algorithm, and $\epsilon$ be as in Equation (9). Then,*

$$\mathbb{H}[j^*, \tilde{n}^*] \leqslant I[\mathbf{X};\mathbf{Z}] + \mathbb{E}_{\mathbf{X}}[\epsilon] + \log_2(I[\mathbf{X};\mathbf{Z}] - \log_2 J + \mathbb{E}_{\mathbf{X}}[\epsilon] + 1) + 6. \tag{100}$$

*Proof.* For those who are not familiar with GPRS, please note the notation used by Flamich (2024) is slightly different from those defined in this paper. We define the sample we encode as $\mathbf{z}^*$, while in GPRS paper, it is denoted by $\tilde{X}$. We define the smallest $\tau$ as $\tau^*$, while in GPRS paper, Flamich (2024) name it as the *first arrival* and denote it by $\tilde{T}$. We define the global sample index of the encoded sample by $n^*$, while Flamich (2024) denote it by $N$. In the following proof, since we will mainly modify the proof of Flamich (2024), we follow their notations unless otherwise stated.

We still follow the structure similar to the proof in Appendix C.1. Specifically, we first derive the upper bound of $\log_2 n^*$ conditional on a certain $\mathbf{X} = \mathbf{x}$. Then, we take expectation over $\mathbf{X}$. Finally, we derive the entropy by the same maximum entropy argument.

**Derive $\mathbb{E}[\log_2 n^*|\mathbf{X} = \mathbf{x}]$.** Our proof starts from Equation (59) on Page 17 of the GPRS paper. It says that $N - 1$ follows a Poisson distribution given $\tilde{T} = t$, with mean:

$$t - \tilde{\mu}(t) \tag{101}$$

where $\tilde{\mu}$ is defined by Flamich (2024) as the average number of points "under the graph" up to time $t$ (See Appendix A on Page 13 of the GPRS paper for a detailed definition).

WLG, assume the encoded sample $\tilde{X}$ is in the $j^*$-th bin. Then, according to the property of Poisson distribution, $n^* - 1$ is also Poisson-distributed, with mean $\pi(j^*) \cdot (t - \tilde{\mu}(t))$, where $n^*$ is the local sample index in the $j^*$-th bin.

Therefore, we can modify Equation (133) on Page 21 of the GPRS paper as follows:

$$\mathbb{E}\big[\log_2 n^*|\tilde{T} = t, \tilde{X} = x, x \in B_{j*}, \mathbf{X} = \mathbf{x}\big] \leqslant \log_2(\mathbb{E}[(n^* - 1)|\tilde{T} = t, \tilde{X} = x, x \in B_{j*}] + 1) \tag{102}$$

$$= \log_2[\pi(j^*) \cdot (t - \tilde{\mu}(t)) + 1] \tag{103}$$

$$\leqslant \log_2[\pi(j^*)t + 1] \tag{104}$$

By Equation (132) on Page 21 of the GPRS paper, we have

$$t \leqslant \frac{q(x)}{p'(x)\mathbb{P}[\tilde{T} \geqslant t]} \tag{105}$$

where $p'$ is the density of our adjusted prior, defined in Equation (25).

Therefore,

$$\pi(j^*)t \leqslant \frac{\pi(j^*)q(x)}{p'(x)\mathbb{P}[\tilde{T} \geqslant t]} \tag{106}$$

$$= \frac{\cancel{\pi(j^*)}P(B_{j*})q(x)}{\cancel{\pi(j^*)}p(x)\mathbb{P}[\tilde{T} \geqslant t]} \tag{107}$$

$$= \frac{q(x)}{J \cdot p(x)\mathbb{P}[\tilde{T} \geqslant t]} \tag{108}$$

Hence, we have

$$\pi(j^*)t + 1 \leqslant \frac{q(x)}{J \cdot p(x)\mathbb{P}[\tilde{T} \geqslant t]} + 1 \tag{109}$$

$$= \frac{q(x)/(J \cdot p(x))}{\mathbb{P}[\tilde{T} \geqslant t]} + 1 \tag{110}$$

$$\leqslant \frac{q(x)/(J \cdot p(x)) + 1}{\mathbb{P}[\tilde{T} \geqslant t]} \tag{111}$$

Taking this back to Equation (104), we have

$$\mathbb{E}[\log_2 n^* | \tilde{T} = t, \tilde{X} = x, x \in B_{j*}, \mathbf{X} = \mathbf{x}] \leqslant \log_2\left[\frac{q(x)/(J \cdot p(x)) + 1}{\mathbb{P}[\tilde{T} \geqslant t]}\right] \tag{112}$$

$$= \log_2\left[\frac{q(x)}{J \cdot p(x)} + 1\right] + \tilde{\mu}(t) \cdot \log_2 e, \tag{113}$$

where the last follows the same principle as Equation (137) on Page 21 of the GPRS paper.

Now, by the law of iterated expectations, we find

$$\mathbb{E}[\log_2 \tilde{n}^* | \mathbf{X} = \mathbf{x}] = \mathbb{E}_{j*}\mathbb{E}_{x \sim Q|_{B_{j*}}}\mathbb{E}_t\left[\mathbb{E}[\log_2 n^* | \tilde{T} = t, \tilde{X} = x, x \in B_{j*}]\right] \tag{114}$$

$$\overset{\text{eq. (113)}}{\leqslant} \mathbb{E}_{j*}\mathbb{E}_{x \sim Q|_{B_{j*}}}\mathbb{E}_t\left[\log_2\left[\frac{q(x)}{J \cdot p(x)} + 1\right] + \tilde{\mu}(t) \cdot \log_2 e\right] \tag{115}$$

$$= \mathbb{E}_{x \sim Q}\left[\log_2\left[\frac{q(x)}{J \cdot p(x)} + 1\right]\right] + \log_2 e \cdot \mathbb{E}_t[\tilde{\mu}(t)] \tag{116}$$

According to Equation (64) on Page 17 of the GPRS paper, $\mathbb{E}_t[\tilde{\mu}(t)] = 1$. Therefore, we have

$$\mathbb{E}[\log_2 \tilde{n}^* | \mathbf{X} = \mathbf{x}] \leqslant \mathbb{E}_{x \sim Q}\left[\log_2\left[\frac{q(x)}{J \cdot p(x)} + 1\right]\right] + \log_2 e \tag{117}$$

Surprisingly, *this coincide with Equation (40) in the proof for PFR up to a constant.* Therefore, following exactly the proof for PFR, we have

$$\mathbb{E}[\log_2 \tilde{n}^* | \mathbf{X} = \mathbf{x}] \leqslant D_{\mathrm{KL}}[Q||P] - \log_2 J + \epsilon + 2 \cdot \log_2 e \tag{118}$$

where $\epsilon$ is defined in the same way as the PFR case.

**Derive $\mathbb{E}[\log_2 \tilde{n}^*]$.** Following exactly the proof for PFR, we have

$$\mathbb{E}[\log_2 \tilde{n}^*] \leqslant I[\mathbf{X}; \mathbf{Z}] - \log_2 J + \mathbb{E}_{\mathbf{X}}[\epsilon] + 2 \cdot \log_2 e \tag{119}$$

where $\epsilon$ is defined in the same way as the PFR case.

**Derive entropy.** Finally, following the same maximum entropy argument, we have

$$\mathbb{H}[\tilde{n}^*] \leqslant I[\mathbf{X}; \mathbf{Z}] - \log_2 J + \mathbb{E}_{\mathbf{X}}[\epsilon] + 2 \cdot \log_2 e + \log_2(I[\mathbf{X}; \mathbf{Z}] - \log_2 J + \mathbb{E}_{\mathbf{X}}[\epsilon] + 2 \cdot \log_2 e + 1) + 1 \tag{120}$$

$$\leqslant I[\mathbf{X}; \mathbf{Z}] - \log_2 J + \mathbb{E}_{\mathbf{X}}[\epsilon] + \log_2(I[\mathbf{X}; \mathbf{Z}] - \log_2 J + \mathbb{E}_{\mathbf{X}}[\epsilon] + 1) + 1 + 2 \cdot \log_2 e + \log_2(2 \cdot \log_2 e + 1) \tag{121}$$

$$\leqslant I[\mathbf{X}; \mathbf{Z}] - \log_2 J + \mathbb{E}_{\mathbf{X}}[\epsilon] + \log_2(I[\mathbf{X}; \mathbf{Z}] - \log_2 J + \mathbb{E}_{\mathbf{X}}[\epsilon] + 1) + 6 \tag{122}$$

Do not forget we need $\log_2 J$ bits to encode the index $j^*$. Therefore, the two-part code's codelength is upper-bounded by

$$\mathbb{H}[\tilde{n}^*] + \log_2 J \leqslant I[\mathbf{X}; \mathbf{Z}] + \mathbb{E}_{\mathbf{X}}[\epsilon] + \log_2(I[\mathbf{X}; \mathbf{Z}] - \log_2 J + \mathbb{E}_{\mathbf{X}}[\epsilon] + 1) + 6 \tag{123}$$

which finishes the proof. $\qquad\square$

## C.4 Proof of Corollary 4.1

**Corollary 4.1.** *Given a target distribution $Q$ and a prior distribution $P$ running Ordered random coding (ORC) to encode a sample from $Q$. Let $\tilde{Q}$ be the distribution of encoded samples. If the number of candidates is $N = 2^{D_{KL}[Q||P]+t}$ for some $t \geq 0$, then*

$$D_{TV}[\tilde{Q}, Q] \leq 4 \left( 2^{-t/4} + 2\sqrt{\mathbb{P}_{\mathbf{z} \sim Q} \left( \log \frac{q(\mathbf{z})}{p(\mathbf{z})} \geq D_{KL}[Q||P] + \frac{t}{2} \right)} \right)^{1/2} \tag{124}$$

*Conversely, suppose that $N = 2^{D_{KL}[Q||P]-t}$ for some $t \geq 0$, then*

$$D_{TV}[\tilde{Q}, Q] \geq 1 - 2^{-t/2} - \mathbb{P}_{\mathbf{z} \sim Q} \left( \log \frac{q(\mathbf{z})}{p(\mathbf{z})} \leq D_{KL}[Q||P] - \frac{t}{2} \right) \tag{125}$$

*Proof.* Theis and Yosri (2022) prove Equation (124). Therefore, we focus on Equation (125).

First, we note that, as proved by Theis and Yosri (2022), the sample returned by ORC has the same distribution as the sample returned by Minimal random coding Havasi et al. (2019) and hence inherits all the conclusions of importance sampler by Chatterjee and Diaconis (2018).

Then, we define set $\mathcal{A}$ as

$$\mathcal{A} = \left\{ \mathbf{z} \,\middle|\, \log_2 \frac{q(\mathbf{z})}{p(\mathbf{z})} \leq D_{\text{KL}}[Q||P] - \frac{t}{2} \right\} \tag{126}$$

also, define

$$f(\mathbf{z}) = \begin{cases} 1 & \mathbf{z} \in \mathcal{A} \\ 0 & \text{otherwise} \end{cases} \tag{127}$$

Note, that

$$\mathbb{E}_Q[f(\mathbf{z})] = \mathbb{P}_{\mathbf{z} \sim Q}[\mathbf{z} \in \mathcal{A}] = Q(\mathcal{A}). \tag{128}$$

Assume given a random state $S$, the set of all candidate samples $\{\mathbf{z}_1, ..., \mathbf{z}_N\}$ is uniquely determined. We therefore denote

$$I_{N,S} = \frac{\sum_{i=1}^N f(\mathbf{z}_i) \frac{q(\mathbf{z}_i)}{p(\mathbf{z}_i)}}{\sum_{i=1}^N \frac{q(\mathbf{z}_i)}{p(\mathbf{z}_i)}} \tag{129}$$

According to Chatterjee and Diaconis (2018), we have

$$\mathbb{E}_S[\mathbb{1}\{I_{N,S} \neq 1\}] = \mathbb{P}[I_{N,S} \neq 1] \leq 2^{-t/2}; \tag{130}$$

$$\mathbb{E}_S[\mathbb{1}\{I_{N,S} = 1\}] = \mathbb{P}[I_{N,S} = 1] \geq 1 - 2^{-t/2}. \tag{131}$$

Therefore, we have

$$\tilde{Q}(\mathcal{A}) = \mathbb{E}_{\tilde{Q}}[f] \tag{132}$$

$$= \mathbb{E}_S[I_{N,S}] \tag{133}$$

$$= \mathbb{P}[I_{N,S} = 1]\mathbb{E}_S[I_{N,S}|I_{N,S} = 1] + \mathbb{P}[I_{N,S} \neq 1]\mathbb{E}_S[I_{N,S}|I_{N,S} \neq 1] \tag{134}$$

$$\geq \mathbb{P}[I_{N,S} = 1]\mathbb{E}_S[I_{N,S}|I_{N,S} = 1] \tag{135}$$

$$= \mathbb{P}[I_{N,S} = 1] \tag{136}$$

$$\geq 1 - 2^{-t/2}. \tag{137}$$

on the other hand, we have

$$Q(\mathcal{A}) = \mathbb{P}_{\mathbf{z} \sim Q}[\mathbf{z} \in \mathcal{A}] = \mathbb{P}_{\mathbf{z} \sim Q} \left( \log \frac{q(\mathbf{z})}{p(\mathbf{z})} \leq D_{\text{KL}}[Q||P] - \frac{t}{2} \right) \tag{138}$$

Therefore, we have

$$D_{\text{TV}}[\tilde{Q}, Q] := \sup_{\mathcal{A}'} |\tilde{Q}(\mathcal{A}') - Q(\mathcal{A}')| \tag{139}$$

$$\geq |\tilde{Q}(\mathcal{A}) - Q(\mathcal{A})| \tag{140}$$

$$\geq \tilde{Q}(\mathcal{A}) - Q(\mathcal{A}) \tag{141}$$

$$\geq 1 - 2^{-t/2} - \mathbb{P}_{\mathbf{z} \sim Q} \left( \log \frac{q(\mathbf{z})}{p(\mathbf{z})} \leq D_{\text{KL}}[Q||P] - \frac{t}{2} \right) \tag{142}$$

$$\square$$

## D  Experiment Details

### D.1  Synthetic Gaussian Toy Examples

#### D.1.1  Generation of the toy examples

We explore the effectiveness of our proposed algorithm on 5D synthetic Gaussian toy examples. Specifically, we assume $p(\mathbf{x}) = \mathcal{N}(\mathbf{x}|0, \text{diag}(\boldsymbol{\sigma}^2))$, and $q(\mathbf{z}|\mathbf{x}) \sim \mathcal{N}(\mathbf{z}|\mathbf{x}, \text{diag}(\boldsymbol{\rho}^2))$. The density of prior $P$ is given by $p(\mathbf{z}) = \mathcal{N}(0, \text{diag}(\boldsymbol{\sigma}^2 + \boldsymbol{\rho}^2))$. Therefore, for a certain realization of $\boldsymbol{\sigma}, \boldsymbol{\rho}$, we have the following REC task: transmitting a sample following $Q$ whose density is $q(\mathbf{z}|\mathbf{x}) \sim \mathcal{N}(\mathbf{z}|\mathbf{x}, \text{diag}(\boldsymbol{\rho}^2))$ with the help of the prior $P$ with density $p(\mathbf{z}) = \mathcal{N}(0, \text{diag}(\boldsymbol{\sigma}^2 + \boldsymbol{\rho}^2))$. To generate multiple toy examples, we randomly draw $\boldsymbol{\sigma}, \boldsymbol{\rho} \sim \mathcal{U}(0, 1)^5$. The dimension-wise mutual information $I[\mathbf{X}, \mathbf{Z}]$ is $\frac{1}{2} \log_2 \left( (\boldsymbol{\sigma}^2 + \boldsymbol{\rho}^2)/\boldsymbol{\rho}^2 \right)$.

#### D.1.2  Space partitioning details

In this toy example, we naively assume the receiver knows both the dimension-wise mutual information and the entire KL divergence between $Q$ and $P$ (but dimension-wise KL is only available to the sender). In practice, we can achieve this by either encoding the KL divergence using negligible bits (for VAE in Section 5) or enforcing the KL budget during optimization (for INRs in Section 5).

Then we partition the $d$-th axis into approximately $2^{\left( \frac{I_d \cdot D_{\text{KL}}[Q \parallel P]}{\Sigma_{d'=1}^{D} I_{d'}} \right)}$ intervals, where $I_d = \frac{1}{2} \log_2 \left( (\sigma_d^2 + \rho_d^2)/\rho_d^2 \right)$ is the mutual information along $d$-th dimension. To achieve this, we adopt the following steps:

- **Initialize**: we first initialize the number of intervals per dimension to 1, i.e., $J^{[d]} \leftarrow 1, \forall d = 1, 2, \cdots, D$; and we also initialize a vector of mutual information $[I_1, I_2, \cdots, I_d]$.

- **Iterate**: we iteratively increase the number of intervals by two in the dimension that currently has the highest mutual information. Specifically, we find $d^* \leftarrow \arg\max_d \{I_1, I_2, \cdots, I_d\}$, and set $J^{[d^*]} \leftarrow 2 J^{[d^*]}$.

- **Update and Repeat**: after increasing $J^{[d^*]}$, we decrement both the mutual information of that dimension and the total KL divergence by 1. We repeat this process until KL is exhausted.

#### D.1.3  Generation of the plots

After this, we run PFR with space partitioning (Algorithm 3) to encode exact samples and ORC with space partitioning (Algorithm 4) for approximate samples. We show the results in Figure 2 and Figure 3, and also include standard PFR and ORC for comparison. To plot Figure 2, we sample 200 different $\boldsymbol{\sigma}, \boldsymbol{\rho}$ pairs, and repeat the encoding procedure 50 times with different seeds for each realization of $\boldsymbol{\sigma}, \boldsymbol{\rho}$. We calculate the negative log-likelihood of the 50 encoded indices using Zipf distribution $P(N) \propto N^{-(1+1/\zeta)}$ as a proxy of the codelength. To find the optimal value of $\zeta$, we optimize the log-likelihood of the 50 indices with `scipy.optimize.minimize`. Finally, we average and find the IQR across these 50 repeats, and smooth the curves at the end for better visualization. In real-world applications, one may raise the concern that we need to transmit this $\zeta$, which may lead to another overhead in the codelength. However, as we will see later in our VAE and INR cases, we can estimate the value of $\zeta$ from the training set (or a very small subset of the training set) and share it between the sender and receiver.

To plot Figure 3, we first sample 200 different $\boldsymbol{\sigma}, \boldsymbol{\rho}$ pairs. For each realization of $\boldsymbol{\sigma}, \boldsymbol{\rho}$, we run ORC with a range of sample sizes: $2^{\{1,2,3,4,5,6,7,8,9,10\}}$, and repeat the encoding procedure 500 times with different seeds for each sample size. Then we estimate maximum mean discrepancy (MMD, Smola et al., 2007) between the 500 encoded samples and real target samples with RBF kernel[5]. Since different target $Q$ can lead to different MMD scales, we normalize the $Q$ to standard isotropic Gaussian (and also scale and shift the encoded samples correspondingly) before calculating MMD. Note, that the absolute value MMD means little here. What we should care about is the relative comparison. After this process, we achieve a scatter plot. Finally, for clear visualization, we fit a

---

[5]We use the codes at https://github.com/jindongwang/transferlearning/blob/master/code/distance/mmd_numpy_sklearn.py (MIT License)

Gaussian Process regressor with `scikit-learn` package (Pedregosa et al., 2011) to estimate the mean and IQR.

## D.2 Lossless Compression on MNIST with VAE

### D.2.1 VAE architecture and training details

We follow (Flamich et al., 2024; Townsend et al., 2019) for the VAE architecture. Specifically, both encoder and decoder are 2-layer MLP. We set the hidden size to 400 and the latent size to 100, with ReLU activation function. We use fully factorized Gaussian as the latent distribution and Beta-Binomial as the likelihood distribution. The encoder maps the input flattened image to two 100-D vectors, corresponding to the mean and std in Gaussian distribution. We use softplus to ensure the standard deviation vector is positive. The decoder maps the 100-D latent vector back to two 784-D vectors, corresponding to two parameters in Beta-Binomial distribution. We also use softplus to ensure positivity. We use Adam (Kingma and Ba, 2017) with a learning rate of 0.001 as the optimizer, to train the VAE with a batch size of 1,000 for 1,000 epochs.

### D.2.2 Space partitioning details

We estimate the mutual information $I_d$ along each dimension $d$ in the latent space (i.e., averaging over the KL divergence across the entire training set), and then we partition the $d$-th axis into approximately $2^{\left(\frac{I_d \cdot \lfloor D_{\mathrm{KL}}[Q \parallel P] \rfloor}{\Sigma_{d'=1}^{D} I_{d'}}\right)}$ intervals. We use the same strategy as Appendix D.1.2, but also found simply rounding $2^{\left(\frac{I_d \cdot \lfloor D_{\mathrm{KL}}[Q \parallel P] \rfloor}{\Sigma_{d'=1}^{D} I_{d'}}\right)}$ to the nearest integer works very well. Here, we apply the floor function to the KL divergence since we need to transmit this value to the receiver.

To ensure our space partitioning strategy can reduce the runtime, we need mutual information to match well with the KL divergence for each test image along all dimensions. We empirically check this in Figure 10. As we can see, the test image KL divergence concentrated around mutual information. Also, if the mutual information is zero for some dimension, the test KL divergence will also be zero.

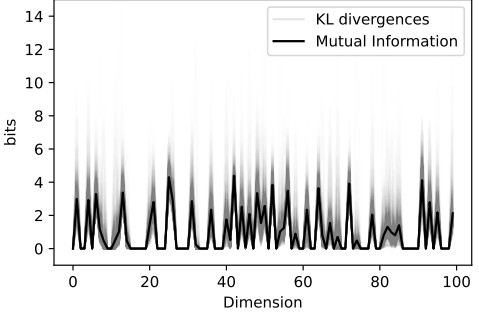

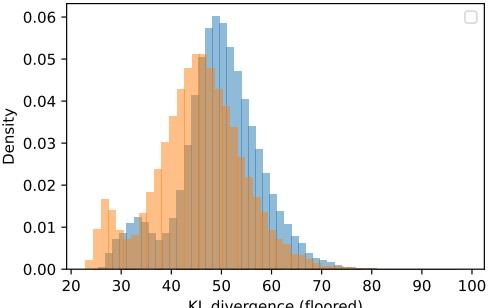

Figure 10: Dimension-wise mutual information and KL divergence for each test image along all dimensions. We can see the KL divergences concentrate around mutual information. And if the mutual information is zero for some dimension, the test KL divergence will also be zero. This ensures our space partitioning strategy can reduce the runtime.

Figure 11: The distribution of KL for each group, estimated from 60,000 training images. Here, we take the case where we split the latent embeddings into 2 blocks as an example. Different colors represent different blocks. We note that the difference between these 2 blocks is solely due to randomness in the splitting.

### D.2.3 Encoding details

We losslessly encode a test image with a code with four components: (1) $\lfloor D_{\mathrm{KL}}[Q \parallel P] \rfloor$ for each block; (2) the local sample index for each block; (3) the bin index for each block and (4) the image conditional on the encoded latent embedding. We discuss these 4 components in the following:

- For the first component, we estimate the distribution of $\lfloor D_{\mathrm{KL}}[Q \parallel P] \rfloor$ for each block from the training set, and use the negative log-likelihood as a proxy for the codelength. To provide an intuition on the distribution of $\lfloor D_{\mathrm{KL}}[Q \parallel P] \rfloor$, we take the 2-block case as an example and visualize the histogram of $\lfloor D_{\mathrm{KL}}[Q \parallel P] \rfloor$ for each block in Figure 11.

- For the second component, we estimate the codelength by the negative log-likelihood of Zipf distribution $P(N) \propto N^{-(1+1/\zeta)}$. We find the optimal $\zeta$ value for each block from only 50 training images and find it generalizes very well to all test images. Note, that we find a different $\zeta$ for each block. This is because, as we can see from Figure 11, the random splitting will not ensure the two blocks to have the same KL divergence distribution and hence may lead to different optimal $\zeta$.

- The codelength for the bin index is simply $\lfloor D_{\mathrm{KL}}[Q \parallel P] \rfloor$ for each block.

- For the last component, we estimate the codelength by the negative log-likelihood of Beta-Binomal distribution with the parameters predicted by the decoder.

One concern is that using negative log-likelihood as a proxy may underestimate the codelength. However, since we use the same proxy across all settings, this doesn't raise an issue in comparison.

### D.3 Lossy Compression on CIFAR-10 with INRs

Following the methodology described by He et al. (2023), we model the latent weights and latent positional encodings (collectively denoted by $\mathbf{h}$) with a fully factorized Gaussian distribution $q_{\mathbf{h}}$. We also learn the prior $p_{\mathbf{h}}$ using 15,000 images from the CIFAR-10 training set. During the testing and encoding stage, He et al. (2023) suggested using 16-bit blocks. Specifically, He et al. (2023) split $\mathbf{h}$ into $K$ blocks, $\mathbf{h} = [\mathbf{h}^{[1]}, \cdots, \mathbf{h}^{[K]}]$, and trained $q_{\mathbf{h}}$ for each test image $\mathbf{x}$ by optimizing the following $\beta$-ELBO:

$$\mathcal{L} = \sum_{k}^{K} \beta_k \cdot D_{\mathrm{KL}}\left[q_{\mathbf{h}^{[k]}} || p_{\mathbf{h}^{[k]}}\right] + \mathbb{E}_{q_{\mathbf{h}}}\left[\mathrm{Distortion}(\hat{\mathbf{x}}_{\mathbf{h}}, \mathbf{x})\right] \tag{143}$$

where $\hat{\mathbf{x}}_{\mathbf{h}}$ is the reconstructed image with parameter $\mathbf{h}$, and $\beta_k$-s are adjusted dynamically to ensure $D_{\mathrm{KL}}\left[q_{\mathbf{h}^{[k]}} || p_{\mathbf{h}^{[k]}}\right] \approx 16$. In our experiments, we find that our proposed methods can safely accommodate larger block sizes of 48 bits. To ensure a fair comparison, we follow He et al. (2023)'s setting on CIFAR-10 exactly, except for the block sizes and the REC algorithm. Specifically, we modify the block sizes and the REC algorithm in codes at `https://github.com/cambridge-mlg/RECOMBINER` (MIT License), and keep other settings unchanged.

As for the theoretical RD curve in Figure 4, we do not split $\mathbf{h}$ into blocks, and directly optimize

$$\mathcal{L} = \beta \cdot D_{\mathrm{KL}}\left[q_{\mathbf{h}} || p_{\mathbf{h}}\right] + \mathbb{E}_{q_{\mathbf{h}}}\left[\mathrm{Distortion}(\hat{\mathbf{x}}_{\mathbf{h}}, \mathbf{x})\right]. \tag{144}$$

The $\beta$ is set to the $\beta$ obtained during training and kept the same for all test samples since we do not need to ensure any bit budget for a theoretically ideal REC codec. Then we directly draw a sample from $q_{\mathbf{h}}$ to calculate PSNR, since we assume a theoretically ideal REC codec can encode a sample from the target exactly. As for the rate, we average $D_{\mathrm{KL}}\left[q_{\mathbf{h}} || p_{\mathbf{h}}\right]$ across all test images to estimate $I[\mathbf{h}, \mathbf{X}]$, and calculate the codelength by $I[\mathbf{h}, \mathbf{X}] + \log_2(I[\mathbf{h}, \mathbf{X}] + 1) + 4$.

#### D.3.1 Space partitioning details

To apply our proposed REC algorithm, we also need to estimate the mutual information $I_d$ along each dimension $d$ in $\mathbf{h}$. To achieve this, we randomly select 1,000 training images, and learn their posterior distributions $q_{\mathbf{h}}$ by Equation (143). We also adjust $\beta_k$ to ensure $D_{\mathrm{KL}}\left[q_{\mathbf{h}^{[k]}} || p_{\mathbf{h}^{[k]}}\right] \approx 48$ on these 1,000 training images. After this, we average their KL divergence per dimension as an estimation of the dimension-wise mutual information. Having the estimation of $I_d$, we use the same strategy as Appendix D.1.2 to partition the space during test time.

#### D.3.2 Encoding details

During the encoding stage, we simply apply Algorithm 4 to encode each block. We use Zipf distribution $P(N) \propto N^{-(1+1/\zeta)}$ to encode the local sample index, and then spend 48 bits to encode the bin index. We find the optimal $\zeta$ value from only 10 training images. Different from the VAE case, we find using the same $\zeta$ value for all blocks works well. Also, to ensure a fair comparison with other codecs, we do not estimate the codelength by log-likelihood but encode the index using `torchac` package (Mentzer et al., 2019) and count the length of the obtained bitstream.

# E Licenses

Datasets:
CIFAR-10 (Krizhevsky et al., 2009): MIT License
MNIST (LeCun and Cortes, 1998): CC BY-SA 3.0 License

Codes and Packages:
Codes for RECOMBINER[6] (He et al., 2023): MIT License
Codes for calculating MMD[7]: MIT License
torchac[8] (Mentzer et al., 2019): GPL-3.0 license

---

[6]https://github.com/cambridge-mlg/RECOMBINER
[7]https://github.com/jindongwang/transferlearning/blob/master/code/distance/mmd_numpy_sklearn.py
[8]https://github.com/fab-jul/torchac

# F More Results

In Figure 12, we compare RECOMBINER (He et al., 2023) with our proposed algorithm with its original performance and also present the RD curves of other codecs for reference. Specifically, our baselines include classical codecs: (JVET, 2020), BPG (Bellard, 2014), JPEG2000; VAE-based codecs: Ballé et al. (2018), Cheng et al. (2020); and INR-based codecs: COIN++ (Dupont et al., 2022), VC-INR (Schwarz et al., 2023) and COMBINER (Guo et al., 2024).

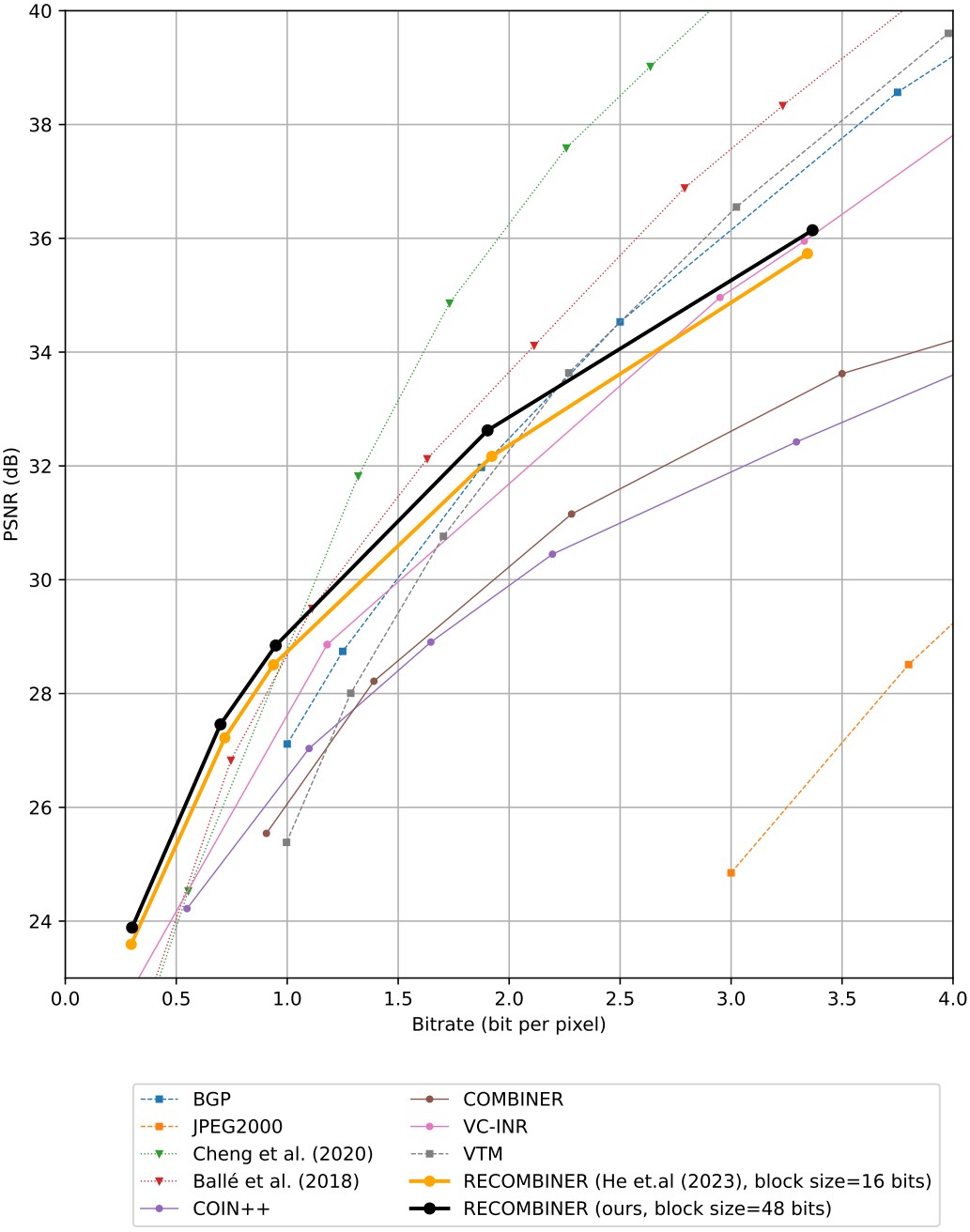

Figure 12: Comparing RECOMBINER with our proposed space partitioning algorithm (w. fine-tuning) with other codecs on CIFAR-10. We use solid lines to denote INR-based codecs, dotted lines to denote VAE-based codecs, and dashed lines to denote classical codecs.

