# OpenReview forum: "Accelerating Relative Entropy Coding with Space Partitioning"
_NeurIPS.cc/2024/Conference — NeurIPS 2024 poster_

### Official Review · Reviewer_wyos · 2024-06-17

**Soundness:** 4
**Presentation:** 4
**Contribution:** 3
**Rating:** 7
**Confidence:** 3

**Summary:**

The authors provide a formalization on how to introduce search heuristics for channel-simulation (sometimes called "relative entropy coding" in the ML literature). The encoder and decoder agree a priori on a binning scheme that divides the support of the prior/public distribution, which is used to control the search during encoding.

Disclaimer: while I'm familiar with the basic literature (e.g. REC, PFRL, ORC), I haven't kept up with the literature since 2022.

I'm more than willing to change my scores if the authors point out I missed something or didn't fully understand the contributions.

**Strengths:**

- The paper is nicely written. This is notably a complicated topic to understand and I feel the authors made quite a didactic effort.
- The excess code-length is bounded for the distributions found in practice for NTC.

**Weaknesses:**

- Line 21: I feel this statement is misleading. In the first scenario (lines 14 to 20), the actual value taken on by the random variable is what is being transmitted. Meanwhile, in channel-simulation, all we care about is that the received quantity be distributed according to $Q_{Z | X}$. These are fundamentally different problems. The comparison would make some sense if, in the first scenario, the decoder is necessarily stochastic, as decoding can be seen as sampling from some posterior distribution over the data conditioned on the latent.

**Questions:**

- Line 66: Why is it important for the state $S$ to be random? I don't see this being used anywhere.

- Line 213: The non-exactness here is with respect to $Q$. REC-like algorithms are usually formulated to be used in the latent space, i.e., the encoder wants to send a sample from $Q_{Z | X}$, but what we care a about from a practical standpoint is to transmit the data $X$ according to some rate-distortion trade-off. Shouldn't this discussion be formulated around the data space (maybe this is too hard of a problem?)?

- Regarding the binning prior $\pi$. The target distribution is a function of $X$, i.e., $Q_{Z | X}$. Isn't it hard to pick a good $\pi$ given this distribution is constantly changing?

**Limitations:**

- Line 267: The results are computationally intensive, and thus impractical, even for MNIST. Requiring the value of mutual information to be known by encoder/decoder seems a bit much (the authors mention this in the limitations section).

---

> ### Author Rebuttal · Authors · 2024-08-05
>
> We thank the reviewer for their detailed and insightful review. We are delighted that the reviewer recognizes the difficulty of this task and appreciates the contribution of our work. Below, we respond to the reviewer's concerns and questions. Should the reviewer find our answers satisfactory, we kindly invite them to consider raising their score.
>
> **Weakness: Line 21 in the Introduction is misleading, quantization and REC are fundamentally different**
>
> We thank the reviewer for raising this concern. Upon revisiting the paragraphs the reviewer mentions, we agree that our argument can easily appear as an apples-to-oranges comparison. Hence, to clarify: the point we wanted to make here is that quantization makes end-to-end learning difficult due to its non-differentiability. Hence, we advocate for REC, which swaps out quantization for some continuous perturbation as a means of limiting the latent representation’s information content, as the latter integrates well with end-to-end training thanks to the reparameterization trick. We will clarify the introduction in our camera-ready version based on this discussion.
>
> However, we’d also like to note that the difference between using 1) quantization + entropy coding or 2) relative entropy coding for lossy compression is not necessarily as big as the reviewer is claiming. The analysis of quantization often relies on treating the quantization error as uniform noise. This approximation has been remarkably successful in the study of quantization and in practice, as it is the foundation of Balle et al.’s neural compression methods as well. Furthermore, Agustsson and Theis (2020) show that quantization-based schemes can be turned into relative entropy coding-based schemes (with uniform target distributions) with a simple modification.
>
> **Regarding Questions:**
>
> 1) > Why is it important for the state $S$ to be random?
>
> Thanks for raising this point. By random state $S$, we mean the common randomness between communication parties, a standard requirement by all channel simulation schemes. We will clarify our terminology in the camera-ready version of the paper.
>
> 2) > Shouldn't this discussion be formulated around the data space?
>
> This is a great question! While we do not discuss this in our paper, this concern is precisely why we use total variation to measure the approximation error, as bounding the error in latent space gives us an immediate guarantee in data space. More formally, denote the decoder by $f$, the true target distribution as $Q$ and the approximate target $\tilde{Q}$. Furthermore, assume $D_{TV}[Q \Vert \tilde{Q}] \leq \epsilon$. By the data processing inequality, we have $D_{TV}[f_{\*}Q  \Vert f_{\*}\tilde{Q}] \leq\epsilon$, where $f_{\*}Q$ denotes the pushforward measure. For more discussion on this topic, please Section III in Flamich and Wells (2024); we will update the camera-ready version to include this discussion.
>
> 3) > Isn't it hard to pick a good $\pi$ given this distribution is constantly changing?
>
> No, thankfully is not difficult!
> - first, the partition is fixed across all data points, and hence, this operation is amortized and not expensive;
>
> - second, for ORC (Sec 4.2), to sample from eq (15), we only need to first sample from $Q$ and find the bin in which the sample lies. Therefore, we do not need extra operations for pre-processing; for PFR (Sec 4.1), we need to calculate $\pi$ in eq(12) per dimension for each data point before sampling. But this is still much cheaper compared to sampling;
>
> - additionally, we highlight that the receiver does not need to be aware of $\pi$ to reconstruct the sample. Therefore, while the $\pi$ we use differs for different data points, there is no extra cost in transmitting.
>
> **Regarding Limitations:**
>
> > The results are computationally intensive, and thus impractical, even for MNIST.
>
>  We do not necessarily agree that the VAE results are computationally intensive and impractical. We use $2^{16}$ samples to encode a block that requires around $2^{50}$ samples by previous methods. With our method, we have already largely reduced the computation cost and made REC more practical, though we definitely believe that this can be further improved.
>
> > Requiring the value of mutual information to be known by encoder/decoder seems a bit much.
>
> We agree that this is not a totally general assumption. However, it is not too much for a practical neural compression application. In most neural compression applications, including data compression and (potentially) federated learning, we have access to training data on which we can easily estimate the dimensionwise mutual information. Also, compared with assumptions for previous faster methods (either assuming 1D or uniform/Gaussian with a known scale per dim), our assumption is already a lot more general.
>
>
>
> **References**
>
> Agustsson, E., & Theis, L. (2020). Universally quantized neural compression. Advances in neural information processing systems, 33, 12367-12376.
>
> Flamich, G., & Wells, L. (2024). Some Notes on the Sample Complexity of Approximate Channel Simulation. arXiv preprint arXiv:2405.04363.

---

> > ### Comment · Reviewer_wyos · 2024-08-09
> >
> > I'm satisfied with the responses in general and have updated my score accordingly. Thank you for the detailed response.
> >
> > > However, we’d also like to note that the difference between using [...]
> >
> > Agreed. To be clear, I in no way was trying to make a technical distinction at the practical level, but just thought it might be easier for a reader outside the field to digest.
> >
> > > We do not necessarily agree that the VAE results are computationally intensive and impractical. [...]
> >
> > This is a fair point: within the REC family this paper significantly improves upon the number of candidates needed, which relates to computational complexity. I guess my comment was geared towards REC methods in general, but I believe that isn't a fair way to review a paper.
> >
> > > We agree that this is not a totally general assumption. [...]
> >
> > I guess this makes sense. It would be nice if this discussion was in the paper, even if in the appendix (but referenced in the main text), as it seems a bit arbitrary at first glance. Maybe an example where this is already assumed to be true, even if implicitly by calculating, for example, cross-correlations in the gaussian setting.

---

> ### Author Response · Authors · 2024-08-09
> **Thank you for your review! Please consider our response**
>
> We thank the reviewer once again for their effort in the reviewing process. As there are only a few working days left in the discussion period, we would like to ask if our response has satisfied the reviewer’s concerns. If so, we kindly invite the reviewer to consider raising their score. If any concerns remain, we are happy to discuss them further here.

---

### Official Review · Reviewer_we4Y · 2024-07-12

**Soundness:** 3
**Presentation:** 4
**Contribution:** 2
**Rating:** 8
**Confidence:** 2

**Summary:**

**Global disclaimer:** I am very unfamiliar with the topic of the paper. I did my best to try and read the literature and understand as much as I could but my input may be very limited.

**Summary:**
The paper focus on relative entropy coding (REC) algorithms and propose to circumvent a major pitfall which is the prohibitive encoding time. Indeed the commonly used PFR algorithm requires to sample points until finding a good enough alignment with the target distribution. Efficient algorithms only exist in dimension 1. The authors propose to refine the algorithm by partitioning the search space into bins that will better drive the sampling from the coding to the target distribution.
The author then theoretically prove that their approach by is founded (Theorem 3.1), and derive precise codelength bounds for the space partition algorithm. They also discuss a generic partitionning strategy.
Finally, a benchmark highlight the benefits of the space partition approach for compression against the standard PFR algorithm on toy example and real datasets.

**Strengths:**

The paper is very didactic, well written and is quite nice to read. The proposed approached is well structured and detailed (sections 3.1). The theory exposed in the paper is sound and the proof seem correct to me.
The benchmark is well made and honest.
Finally the appendix is very helpful, I particularly appreciated appendix B that shed some light on the partition strategy.

**Weaknesses:**

The idea is finally quite elementary: dividing the space once for all into bins. The proofs are "elementary" (still technical) refinements from the classical PFR theory (starting point is often [Li and El Gamal, 2018] and [ Flamich, 2024]).

**Questions:**

- How the bounds from Theorem 3.1 do compare with the bound from [Li and El Gamal, 2018, Theorem 1] in the same setting, i.e. when $J=1$? More generally, could we recover a bound such as Proposition 3.2 but with a dependence in $J$.
- The number of intervals for each axis is justified in Appendix B, but could the author give more details about how these interval are constructed?
- The proposed approach to the construction of the bins follow depends on the chosen representation. Would it be possible to automatically find a transformation prior to the bining that would provide better results while keeping the same strategy?
- Is there a way to update $\pi$ along the algorithm, using e.g. the Gumbel trick and gradient based method?
- It would be interesting to add an ablation study on the number of intervals per axis.
- Table 1 shows some improvements in number of steps but it would be interesting to have the same thing in time. Could you elaborate about Figure 3, I do not understand what is meant by runtime here.

**Limitations:**

The approach is not fully generic and despite very good success on some examples, it could fail on other as clearly explained in Appendix B.

Typos: the punctuation missing in the maths of the appendix

**Proposed score:**
My rating is based on the fact that the paper is sound, the maths are correct but the scope of the idea is quite simple. I do believe that this work is well presented, interesting and worth publication. On the other hand I am not familiar with this field. As such, I propose a score of 5/10 (a weak accept) but I am very open to discussion with the authors and the other reviewers.

---

> ### Author Rebuttal · Authors · 2024-08-05
>
> We thank the reviewer for their dedicated time and constructive review, which we feel largely enhances the quality of our paper. While the reviewer may not be familiar with this field, we believe they understand most of our manuscript and method well.
>
> We are delighted that the reviewer found our paper well-written and recognized its soundness. Below, we respond to the reviewer's concerns, and we are happy to discuss any further questions the reviewer might have.
> Should the reviewer find our answers satisfactory, we kindly invite them to consider raising their score.
>
> **Weakness: the idea and proofs are elementary**: While we agree that the main idea behind our method is simple, we respectfully disagree that it is a weakness on three grounds. (a) ideas that are simple and work well in practice have the largest potential to be impactful in the field as they are easy to understand and replicate. This is the case for our method: our ideas improved performance by many orders of magnitude in terms of the largest KL values that many future REC algorithms could consider.  (b) the theory justifying this idea required new insight; mainly around deriving the overhead term $\epsilon$ that previous works did not deal with or consider. (c) the general insight that we can use the dimensionwise mutual information to devise schemes to improve compression speed is a significant contribution to the field, and we believe that any future scheme that does not use this additional information will be either limited in speed (as all previous schemes are) or codelength.
>
> **Regarding Questions:**
>
> 1) > How Theorem 3.1 compare with the bound from [Li and El Gamal, 2018, Theorem 1] when $J=1$
>
> when $J=1$, our algorithm is equivalent to PFR, and therefore, we end up with the same bound as [Li and El Gamal, 2018, Theorem 1].
> This can be seen from eq(38) on page 17: when $J=1$, eq (38) becomes $E\log_2[ \frac{q(z)}{p(z)}+1]$. We can then follow the proof of [Li and El Gamal, 2018, Appendix p13].
>
> > Could we recover a bound with a dependence in $J$?
>
> Please note that $\epsilon$ in Theorem 3.1 already expresses a fine-grained dependence on $J$. Furthermore, Proposition 3.2 also depends on $J$ in the sense that it is derived by the extreme case $J = 2^{KL}$ and holds for all $J \leq 2^{KL}$. It is an interesting question if there is a bound that interpolates $J=1$ and $J = 2^{KL}$. In our paper, we did not explore this because, for our method, we always want to set $J$ as large as possible to maximize the gain in speed.
>
> 2) > Could the author give more details about how these intervals are constructed?
>
> As we discussed in our manuscript from line 241 to line 249, we partition the d-th axis into approximately $2^{n_d}$ intervals, where $n_d = D_{KL}[Q||P] I_d / \sum_{d’} I_{d’}$. We provide the implementation details of constructing these bins from line 621 to line 630.
>
> In fact, we empirically found that the methods are NOT very sensitive to how we construct the intervals, as long as we avoid partitioning along uninformative dimensions. Please see the rebuttal PDF (Fig 3) for further empirical verification.
>
> 3) > Would it be possible to automatically find a transformation prior to the binning that would provide better results while keeping the same strategy:
>
> It is not possible to design such a transformation. Note that to avoid sending the transformation (which can be expensive), it should be shared across all data points X. Therefore, if we want to find a transformation to the prior, it is equivalent to asking if we can find a better prior that provides better results on average.
>
> 4) > Is there a way to update $\pi$ along the algorithm
>
> Since our algorithm will work for any $\pi$, it is possible to update it. However, there is no point in doing so as the optimal form of $\pi$ is already given in eq (12) and eq (15), which we can use at negligible computational cost.
>
> 5) > It would be interesting to add an ablation study on the number of intervals per axis
>
> Thank you for this suggestion. We added ablation studies on toy examples and INR experiments. Please find the results in the PDF we uploaded to global rebuttal. We will add these plots to our camera-ready version. We now explain these studies, and please also see the global rebuttal for a more detailed description:
>
> - We first provide a qualitative visualization in Figure 1 in our rebuttal comparing 3 different binning strategies: only partitioning the collapsed dimension; randomly assigning the number of intervals per axis; and assigning intervals per axis according to MI (our suggested strategy). We also include standard ORC for reference. As we can see, our suggested strategy works best, whereas partitioning only the collapsed dimension yields almost the same results as standard ORC. This verifies our discussion on how the partitioning strategies will influence the runtime in Appendix B.2 in our manuscript.
>
> - We then provide quantitive ablations on toy examples (Fig 2) and INR experiments (Fig 3). We can see our suggested partition strategy (partitioning according to MI) is always better than randomly assigning intervals per axis. Also, we found if we first remove uninformative dimensions (mutual information $\approx$ 0), and then randomly assign intervals to other dimensions, the performance is only slightly worse than our proposed partition strategy. This indicates that our algorithm is not very sensitive to how we construct the intervals, as long as we avoid partitioning along uninformative dimensions.
>
> 6) > it would be interesting to have Tab 1 in time
>
> The sample size is linearly proportional to the runtime, so we only provide this in Fig 3 and Tab 1. The sample size is a better metric as it is implementation-agnostic, unlike wallclock time.
>
> > I do not understand what is meant by runtime in Fig 3
>
> This figure shows ‘how many samples in ORC we need for a certain bias’. The y-axis shows the sample size to achieve a certain bias (measured by MMD).

---

> ### Author Response · Authors · 2024-08-09
> **Thank you for your review! Please consider our response**
>
> We thank the reviewer once again for their effort in reviewing our paper and for their constructive review. As there are only a few working days left in the discussion period, we would like to ask if our response has satisfied the reviewer’s concerns. If so, we kindly invite the reviewer to consider raising their score. If any concerns remain, we are happy to discuss them further here.

---

> > ### Comment · Reviewer_we4Y · 2024-08-12
> > **Response to authors**
> >
> > I read the response, as well as the other reviews, and took time to understand the author's responses to all raised points. Overall I am pleased with the author's answers, some points I missed during the review have been clarified.
> >
> > **Weakness: the idea and proofs are elementary**
> > I fully agree with the author's response, thank you for the clarifications. I would encourage the author to make point (c) maybe a little bit clearer in the introduction of the paper.
> >
> > I believe the two added figures in Appendix B2 fill an important gap in the original manuscript and shed some light on the proposed approach.
> >
> > I think I understood the paper, the ideas, and the approach the author are proposing and I think this work is strong and worth acceptance.
> >
> > As such I raise my grade to 8.

---

### Official Review · Reviewer_tEuP · 2024-07-13

**Soundness:** 3
**Presentation:** 3
**Contribution:** 3
**Rating:** 7
**Confidence:** 2

**Summary:**

The paper proposes a space partioning technique to speed up  relative entropy coding.
In entropy coding the sender first transforms X into a representation Z that they encode.
This particular step can done  by Poisson functional representation (PFR).
Unfortunately, PFR’s random runtime can be a significant drawback.

To circumvent this issue,  the author propose a space partitioning scheme to reduce runtime in practical scenarios.
This can be seen (sic) as a search heuristic to a standard REC algorithm

The author then proceed to give a theoretical analysis of the method and support their improvement with numerical results.

**Strengths:**

The paper adresses an important practical problem with many important application.
The algorithm and theoretical results are stated clearly.
The context and related work is clear to the reviewer (who is not an expert in the field).
The numerics show clearly the advantage/improvements  yielded by the technique.

**Weaknesses:**

Although the proposed technique is "simple and efficient", it lacks discussion whether the binning technique impacts the final result (besides codelength).

**Questions:**

- it is unclear to the reviewer how the grid should be set.
-it is unclear to the reviewer how the choice of prior for the binning distribution influences the algorithm.
-  the weakness section raises a question that I would like to see adressed:  how is the binning technique impacting the final result (besides codelength).
I agree that the uniform distribution seems like a natural choice, but it would be good to discuss the choice of prior here

I will happily increase my grade if those questions are properly adressed.

**Limitations:**

The authors recognize some clear limitations of their approach in the conclusion section.
In particular, the approach proposed by the authors is too limited and does not extend to all possible practical cases.
The reviewer agrees with those limitations, although still believes they should not be an obstacle for publication. as this work is a "first step" in this direction.

---

> ### Author Rebuttal · Authors · 2024-08-05
>
> We are thankful for the reviewer's time and detailed review and are delighted that the reviewer found our paper clear and appreciated our technical and theoretical contributions.
> Below, we respond to the reviewer’s questions. We are happy to discuss any further concerns the reviewer might have.
> Should we have addressed the reviewer’s concerns and questions, we kindly invite them to raise their score.
>
>
> 1. > it is unclear to the reviewer how the grid should be set:
>
> We discuss how we set the bins in our manuscript from line 241 to line 249; we will signpost this discussion better in the camera-ready version.
>
> In short, in most neural compression applications, we can compute the mutual information per dimension $I_d$ from the training set. Specifically, for each training data $X$, we have access to its posterior $Q_{Z|X}$ during training, and we can calculate the KL divergence between $Q_{Z|X}$ and the prior $P_Z$ along each dimension. The dimension-wise MI $I_d$ is estimated by averaging the dimension-wise KL across all training data.
> Knowing the mutual information per dimension $I_d$, we partition the d-th axis into approximately $2^{n_d}$ intervals, where $n_d = D_{KL}[Q||P]\cdot  I_d / \sum_{d’} I_{d’}$. We discuss the motivation for this choice in Appendix B.2. We also provide the implementation details from line 621 to line 630 on Page 24.
>
>
> 2. > how the binning technique impacts the final result (besides codelength):
>
> In Appendix B.2, we discuss how different binning strategies influence the results. For a further illustration of the effect of the chosen grid on the efficiency of the algorithm, please see Figure 1 in the attached rebuttal PDF.
>
> Essentially, if we only partition the space by dividing the axes where the prior $P$ and the target $Q$ have the same marginal, we will not reduce the runtime. However, luckily, as we discussed above, in most neural compression applications, we can compute the mutual information per dimension $I_d$ on the training set. If $I_d> 0$, it tells us, on average, that the prior $P$ and the target $Q$ will not have the same marginal along the d-th axis. Therefore, we propose to partition the d-th axis into approximately $2^{n_d}$ intervals, where $n_d = D_{KL}[Q||P]\cdot  I_d / \sum_{d’} I_{d’}$.
>
> To help the reviewer further understand this argument, we use a toy example for visualization in our attached rebuttal PDF. In Figure 1 in our rebuttal PDF, we create a toy 5D Gaussian target, and we set dimension 1 to have 0 mutual information (we call it a collapsed/uninformative dimension). We compare our proposed method with 3 different binning strategies: only partitioning the collapsed dimension; randomly assigning the number of intervals per dimension; and partitioning the d-th axis into approximately $2^{n_d}$ intervals, where $n_d = D_{KL}[Q||P] I_d / \sum_{d’} I_{d’}$. We also include standard ORC for reference.
> We run the 4 settings with the same number of samples (20) and repeat each setting 5000 times. The error can be seen from the histogram of 5000 encoded results.
> We can see that partitioning according to mutual information yields the best results, whereas partitioning only the collapsed dimension yields almost the same results as standard ORC.
>
>
> 3. > how the choice of prior for the binning distribution influences the algorithm:
>
> We assume the reviewer is asking how the results will be influenced when each bin has different probabilities under the prior. We answer this question below, and we are happy to address any further questions if we have misunderstood the reviewer's question.
>
> Setting all bins to have the same probability mass under the prior is the basic requirement for our algorithm to adhere to the codelength outlined in Theorem 3.1 and Proposition 3.2. We note this in lines 104 and 414.
>
> This can be understood intuitively: under an ideal Bayesian treatment, the average Q_{Z|X} across all data X should equal the prior $P_Z$. Also, we note that the partitions are shared across all data X in the dataset. Therefore, ideally, we will expect the average probability (average is taken over all randomness and all X) that the encoded sample falls into any bin should be the same.   We can also see the reason from our proof of Theorem 3.1. Specifically, we replace all $P(B_j)$ in eq(37) by $1/J$ in eq(38), which simplifies the expression and is essential for the subsequent derivation.
>
> **In Summary:** our algorithm and theory require all bins to have the same probability under the prior.  Theorem 3.1 and Proposition 3.2 will hold for all binning strategies that satisfy this requirement.
> However, there are many (actually, infinitely many) binning strategies that satisfy this requirement.
> Different choices of binning strategies can have different runtimes or different biases given the same runtime budget. To this end, we propose to partition the d-th axis into approximately $2^{n_d}$ intervals, where $n_d = D_{KL}[Q||P] I_d / \sum_{d’} I_{d’}$. We have discussed the motivation and implementation details in our manuscript and provided an additional visualization to aid understanding in our rebuttal PDF(Fig 1) in the global response.

---

> ### Author Response · Authors · 2024-08-09
> **Thank you for your review! Please consider our response**
>
> We thank the reviewer once again for the effort put into reviewing our paper. As there are only a few working days left in the discussion period, we would like to ask if our response has satisfied the reviewer’s concerns. If so, we kindly invite the reviewer to consider raising their score. If any concerns remain, we are happy to discuss them further here.

---

> > ### Comment · Reviewer_tEuP · 2024-08-10
> > **Thank you for your response.**
> >
> > I'm satisfied with the responses provided by the authors.
> > I'm happy about the clarification and apologize for missing any details.
> > I updated my score to reflect the discussion.

---

### Author Rebuttal · Authors · 2024-08-05

We extend our gratitude to all the reviewers for their detailed and comprehensive reviews and for their time spent reviewing our manuscript. We are delighted that the reviewers found our paper easy to follow and recognized our method's technical and theoretical contributions. We addressed their concerns in our responses.

We also attach a PDF showing additional illustrative and ablation experiments for our method. Namely:

- **In Figure 1, we visualize the approximation error of using ORC with different partitioning strategies on a toy problem**.  we create a 5D Gaussian target, and we set dimension 1 to have 0 mutual information(we call it a collapsed/uninformative dimension).
We compare three partition strategies: only partitioning the collapsed dimension; randomly assigning intervals to dimensions; and assigning intervals according to MI (our proposed strategy; specifically, we partition the d-th axis into approximately $2^{n_d}$ intervals, where $n_d = D_{KL}[Q||P]\cdot I_d / \sum_{d’} I_{d’}$). We also include standard ORC for reference. We run the four settings with the same number of samples (20) and repeat each setting 5000 times. We show the histogram of 5000 encoded results.
*As we can see, assigning intervals according to MI  works best, whereas partitioning only the collapsed dimension yields almost the same results as standard ORC. This verifies our discussion on how the partitioning strategies will influence the runtime and how we should choose the intervals in practice in Appendix B.2 in our manuscript.*



- **As suggested by the reviewers we4Y, we provide ablation studies showing how partition strategies will influence performance**.
We run the ablation on the Gaussian toy example (the experiment described in line 260 in our manuscript) and the CIFAR-10 compression experiments (the experiment described in line 286 in our manuscript).
The results are shown in Figures 2 and 3 in our rebuttal PDF. *As we can see, our proposed partition strategy (partitioning each dimension according to mutual information) is always better than randomly assigning intervals per dim*. For CIFAR-10, we also compare the results by first removing uninformative dimensions (mutual information $\approx$ 0), and then randomly assigning intervals to other dimensions. We find this is only slightly worse than our proposed partition strategy. *This not only further verifies our discussion in Appendix B.2, but also indicates that our algorithm is not very sensitive to how we construct the intervals, as long as we avoid partitioning along uninformative dimensions.*

---

### Decision · Program_Chairs · 2024-09-25

**Decision:**

Accept (poster)

**Comment:**

This paper proposes an heuristic to improve relative entropy coding algorithms, and validates its method numerically.

The reviewers unanimously agreed that this well-written paper effectively addresses important questions: I recommend it for acceptance.